# Long open path measurements of greenhouse gases in air using near infrared Fourier transform spectroscopy

David W. T. Griffith[1], Denis Pöhler[2], Stefan Schmitt[2], Samuel Hammer[2], Sanam N. Vardag[2,3], and Ulrich Platt[2]

[1] Centre for Atmospheric Chemistry, University of Wollongong, Australia
[2] Institute of Environmental Physics, University of Heidelberg, Germany
[3] now at Heidelberg Centre for the Environment, University of Heidelberg, Germany

*Correspondence to*: David Griffith (Griffith@uow.edu.au)

**Abstract**

In complex and urban environments, atmospheric trace gas composition is highly variable in time and space. Point measurement techniques for trace gases with in situ instruments are well established and accurate, but do not provide spatial averaging to compare against developing high resolution atmospheric models of composition and meteorology with resolutions of the order of a kilometre. Open path measurement techniques provide path average concentrations and spatial averaging which, if sufficiently accurate, may be better suited to assessment and interpretation with such models. Open path Fourier Transform Spectroscopy (FTS) in the mid infrared region, and Differential Optical Absorption Spectroscopy (DOAS) in the UV and visible, have been used for many years for open path spectroscopic measurements of selected species in both clean air and in polluted environments. Near infrared instrumentation allows measurements over longer paths than mid infrared FTS for species such as greenhouse gases which are not easily accessible to DOAS.

In this pilot study we present the first open path near infrared (4000-10,000 cm$^{-1}$, 1.0 – 2.5 µm) FTS measurements of $CO_2$, $CH_4$, $O_2$, $H_2O$ and HDO over a 1.5 km path in urban Heidelberg, Germany. We describe the construction of the open path FTS system, the analysis of the collected spectra, several measures of precision and accuracy of the measurements, and the results from a four-month trial measurement period in July-November 2014. The open path measurements are compared to calibrated in situ measurements made at one end of the open path. We observe significant differences of the order of a few ppm for $CO_2$ and a few tens of ppb for $CH_4$ between the open path and point measurements 2-4 times the measurement repeatability, but we cannot unequivocally assign the differences to specific local sources or sinks. We conclude that open path FTS may provide a valuable new tool for investigations of atmospheric trace gas composition in complex, small scale environments such as cities.

**Introduction**

The cycling of carbon between Earth's surface and the atmosphere is dominated by carbon dioxide ($CO_2$) and methane ($CH_4$), which are also the two most important anthropogenically-influenced greenhouse gases. The steady increases in atmospheric $CO_2$ and $CH_4$ concentrations in the global atmosphere since industrialisation have been well documented by the global network of surface in situ point measurements (e.g.GLOBAL-VIEW-$CO_2$, 2009). Such point-based in situ measurements in clean baseline air are well suited to monitoring long term global changes in atmospheric greenhouse gases (including also nitrous oxide ($N_2O$) and other minor species), and have provided most of the data from which long term global trends have been assessed. However to characterise and quantify individual sources and sinks of greenhouse gases, measurements in regional, urban, agricultural and industrial environments located near the sources and sinks, combined with fine-resolution local and regional-scale atmospheric transport modelling, are required. In a recent modelling study, Turner et al. (2016) concluded that a dense (2 km) fixed network of point sensors with only moderate precision was sufficient to characterise $CO_2$ sources with 5% accuracy in the San Francisco Bay area. Lee et al. (2016) trialled a network of five mobile $CO_2$ sensors in the Vancouver urban area combined with an aerodynamic model to calculate fluxes.

Point measurements are sensitive to the immediate local environment, and may or may not adequately represent the mean concentrations over the grid-scale of the relevant atmospheric models in non-background environments. Open path (OP) measurements provide spatially averaged concentrations by measuring an optical absorption spectrum along a path between a light source and the measuring instrument and retrieving component concentrations from the spectra. Spatial averaging at similar scales to those of the finest urban and regional scale models should be advantageous in combining measurements and models to deduce the strengths of localised sources and sinks of greenhouse gases. But how accurately can we measure such spatially averaged trace gas concentrations?

The longest-established surface OP techniques (i.e. excluding satellite and ground based total column measurements) are Differential Optical Absorption Spectroscopy (DOAS), typically employing the UV and visible spectral regions (Platt and Stutz, 2008), and Open Path -Fourier Transform Spectroscopy (OP-FTS) in the mid infrared (e.g. Tuazon et al., 1978; Russwurm and Childers, 2002; Griffith and Jamie, 2006; Smith et al., 2011; Laubach et al., 2013; Flesch et al., 2016; You et al., 2017). While DOAS can operate over pathlengths of several kilometres, suitable absorptions for accurate and precise measurements of $CO_2$, $CH_4$ and other GHGs are not available in the UV-visible spectrum. In the mid IR suitable absorptions are available, but when restricted to conventional broadband blackbody sources such as a globar, low source brightness limits beam collimation across the open path and restricts pathlengths to typically a few hundred metres. Until recently the near infrared (NIR) region has been little used. For broadband studies, the NIR allows the use of a high temperature, bright white light source (such as quartz halogen or Xe lamp) allowing good beam collimation over kilometre-scale pathlengths, but absorption strengths of the available overtone and combination vibrational spectral bands are much lower than for the fundamental transitions in the mid IR. Previous work to extend DOAS into the NIR region using a conventional white-light source, monochromator and detector array was limited by the weak absorptions and interfering spectral structures to a

repeatability of approximately 30% and uncertain accuracy for $CO_2$ and $CH_4$ (Sommer, 2012). More recently, DOAS - type NIR measurements using broadband laser sources (Saito et al., 2015; Somekawa et al., 2011), and frequency comb spectroscopy (Rieker et al., 2014; Waxman et al., 2017) have been described to measure $CO_2$ and $CH_4$ in the NIR over pathlengths of up to 5 km. These methods achieved measurement repeatabilities of 1-4 ppm with absolute bias of up to 7 ppm for $CO_2$ when compared to point in situ measurements. Other recent developments include open path tunable diode laser (TDL) systems (e.g. Dobler et al., 2013; Queisser et al., 2016), and commercially available laser-based open path analysers (e.g. Boreal Laser Inc., Edmonton, Canada). TDL systems are generally applicable only to a single target gas.

The recent and rapid development of TCCON, the Total Carbon Column Observing Network (Wunch et al., 2011) has shown that the near IR spectrum with a ground based FT spectrometer and the sun as a source is suitable for highly accurate and repeatable (<0.2%) measurements of total column $CO_2$, $CH_4$, $N_2O$ and other trace gases. Smith et al., (2011) assessed the performance of OP-FTS in the mid infrared, finding accuracies of a few percent without calibration against standards. In this work, drawing on our combined experience in TCCON, mid IR OP-FTS and DOAS, we describe measurements of $CO_2$, $CH_4$, $H_2O$, HDO, $O_2$ and other gases with a Fourier Transform Spectrometer (FTS) operating in the near infrared (4000-10000 cm$^{-1}$, 1.1 – 2.5 μm) using a simple broad-band tungsten halogen light source combined with a long open path telescope and retro reflector system over a 1.5 km path (one-way, 3.1 km total absorption pathlength) in urban Heidelberg, Germany. The spectroscopy is similar to that used in TCCON, and in this pilot study we aimed to (1) assess the precision, accuracy and stability of such ground based long open path measurements and (2) compare and test for biases between open path measurements and point measurements made with a calibrated in situ analyser at one end of the open path. The measurement system operated for 4 months from July – November 2014 in urban Heidelberg, Germany.

## 2 Experimental

### 2.1 FT Spectrometer and long path optics

The optical system is shown schematically in Figure 1. The spectrometer and telescope were located in the rooftop observatory on the 6-storey Institute of Environmental Physics (IUP) building on the University of Heidelberg campus in urban Heidelberg, (49.4172°N, 8.6745°E, 145 masl, 33 m above ground) and the retroreflector array on the Institute of Physics (PI) building 1555 m east at (49.4149°N, 8.6956°E, 169 masl). The distance was measured with a laser rangefinder to ±1 m. The intervening path is illustrated in Figure 2 and crossed above a residential area approximately 0.5 km north of the Neckar River and 1.5 km NE of the Heidelberg city centre. A 35W tungsten-quartz-halogen light source was focussed by a 25 mm focal length, 25 mm diameter NIR-coated glass lens (Edmund Scientific, not shown) into a 6 x 200 μm fibre bundle (3 m long, 200/240 IRAN, Loptec GmbH) and directed to the primary focus of a 300 mm diameter, 150 cm focal length Newtonian telescope (aluminium primary mirror with $SiO_2$ overcoat). The collimated beam from the telescope was directed via fine step-control alignment motors to an array of 17 x 63 mm diameter solid UV quartz cornercubes which acted as retroreflectors to return the beam to

the telescope. The focussed return beam was collected by a single 200 μm fibre in the centre of the 6-fibre bundle in the same sheath, which forked to direct the single central fibre to the input of the FT spectrometer. The fibre coupling to the telescope is described in detail by Merten et al. (2011). In practice the fibre end at the telescope was slightly defocussed to maximise the light throughput to the spectrometer.

5 The return beam from the fibre was focussed by a 75 mm focal length NIR-coated lens into the 1 mm entrance stop of the FT spectrometer (IRcube, Bruker Optics, Ettlingen Germany) which had a quartz beamsplitter and InGaAs detector optimised for the NIR spectral region (3800 – 10000 $cm^{-1}$). A typical spectrum is shown in Figure 3. The lower frequency cutoff was determined by the transmission of the UV-quartz cornercubes, fibres and detector.

The rms spectral signal : noise ratio (SNR) was determined at 6300-6500 $cm^{-1}$ from the ratio of two successive 5 minute spectra 10 where atmospheric and fibre residual features mostly cancel leaving only the instrument noise. The observed SNR was typically 700-900:1 for such a ratio spectrum, corresponding to 1000-1200 : 1 for a single spectrum.

Measurements reported here were collected continuously from 10 July – 4 November 2014. Spectra were recorded with a resolution of 0.55 $cm^{-1}$ (maximum optical path difference 1.8 cm), each by coadding 84 scans over 5 minutes. Each hour a background stray light spectrum was recorded by blocking the source at the fibre input and a short path reference spectrum 15 was recorded by blocking the beam at the telescope end of the fibre with an aluminium diffuse reflector plate to return a small fraction of the intensity to the detector without traversing the long open path. Over the 4 month measurement period more than 26,000 spectra were collected, of which approx. 3000 (11%) were rejected due to poor visibility and low signal or other, normally weather-related effects. In total, taking into account hourly background spectrum measurements, downtime due to maintenance and extended poor weather periods, we collected and analysed usable data for 68% of the total time from 10 July 20 to 4 Nov.

Atmospheric pressure and temperature for the measurement path are required for the spectrum analysis and to calculate air density, and were measured and averaged over the period of each spectrum measurement by an electronic barometer (Vaisala PTB110) and LM335 diode respectively, co-located with the FT spectrometer. The acquisition of spectral data, pressure and temperature, shutter control and real-time spectrum analysis were executed automatically by the software available for the 25 Ecotech Spectronus in situ FTIR analyser (Ecotech, Knoxfield , Australia). Initially the IUP weather station temperature and height-adjusted pressure were used in the spectrum analysis; the weather station temperature was subsequently replaced by the path-averaged temperature derived from the spectra themselves, as described below.

**2.2 In situ trace gas measurements**

At the IUP end of the open path, air from a roof-level inlet on the IUP building was sampled and analysed continuously with 30 an in situ trace gas analyser described in detail elsewhere (Griffith et al., 2012; Hammer et al., 2013; Vardag et al., 2015). This analyser is based on an FTIR spectrometer operating in the mid-IR and provided simultaneous high precision measurements of $CO_2$, $CH_4$, CO, $N_2O$, $\delta^{13}C$ and $\delta^{18}O$ in $CO_2$ calibrated against WMO-GAW standards and provided calibrated point measurements for comparison with the path averaged open path measurements. The calibration frequency (daily target tank,

weekly calibration tanks) ensured that all measurements meet GAW compatibility requirements. Measurements were made continuously, averaged every 3 minutes, and the time series was interpolated to the mean times of the open path measurements for point-by-point comparison.

*Meteorological measurements*

Standard measurements of pressure, temperature, humidity, wind speed, wind direction and solar radiation were obtained from the IUP weather station located on the roof of the building as 5 minute averages and interpolated to the times of the open path measurements.

## 2.3 Spectrum analysis and retrieval of trace gas amounts

Path averaged trace gas mole fractions were retrieved from spectra by iteratively best-fitting a calculated spectrum to the
10 measured spectrum. The forward model, MALT (Griffith, 1996) calculates the transmission spectrum from a set of input parameters including absorption line parameters, trace gas amounts, pressure, temperature and pathlength as well as instrument parameters including resolution, apodisation function, lineshape, spectral shift and a five-term polynomial fit to the continuum, which in these single beam spectra is generally not flat. The line parameters are based on Hitran 2008 (Rothman et al., 2009) updated by Toon and co-workers for the GFIT software used throughout TCCON (Wunch et al., 2015). The inverse model
uses non-linear least squares following the Levenberg-Marquart algorithm (Press et al., 1992) to retrieve the path averaged concentration of each trace gas species. The path averaged concentrations are converted to mole fractions by dividing by the concentration of air determined from pressure and temperature. More details are given by Griffith et al. (2012).

Details of the spectral windows used for the NIR long path analysis are summarised in Table 1 and typical fits for spectral regions used to retrieve $O_2$, $CO_2$ and $CH_4$ are shown in Figure 4. The weaker bands near 6300 cm$^{-1}$ (1.58 µm) used in total
20 column TCCON analyses were also analysed but are not included because their signal-to-noise ratio (SNR) is much less than that of the stronger 4800-5000 cm$^{-1}$ bands used here and their contribution to an SNR-weighted mean $CO_2$ retrieval is negligible. Note these spectral windows are quite different from those used in the mid-IR in the in situ analyser (Griffith et al., 2012).

The fibre optic coupling between telescope, source and detector introduces repeatable fringing and interferences in the
25 measured spectra at about 1% of the measured signal intensity. These spectral structures can be seen in the residual plots of Figure 4 and are quite reproducible over periods of days to weeks. They are larger than the underlying detector noise but much less than the trace gas absorptions, at least for $CO_2$ and $O_2$ (Figure 4). Removing or co-fitting an average fibre residual spectrum during the fit makes only a small (<<1%) difference to the retrieved mole fractions because the fibre residual spectrum is itself derived from the least squares fits to real spectra and is approximately orthogonal to the target gas spectrum.

Background spectra of stray light measured hourly by blocking the source had intensities up to 1% of those of the open path spectra, maximising in the early morning and late evening when the solar elevation was low and direction roughly parallel (E-

W) to the open path. Scattered solar stray light collected by the FTIR spectrometer has an effective atmospheric path of >8 km depending on zenith angle, leading to stronger path-average trace gas absorption and higher apparent column amounts of trace gases retrieved from the spectra – for $CO_2$ the enhancement can be up to 5 ppm at low sun elevations with an additional spike apparent when the near-direct solar beam is captured (see example for $O_2$ below). The enhancement is typically less than 1-2 ppm during the middle of the day and at night. The analyses were not corrected for stray light because (a) the stray light spectra were measured only once per hour so they do not provide an accurate measurement of the scattered light at the time of each 5 minute OP measurement, and (b) the stray light spectrum is weak and noisy and adds noise to the retrieved trace gas amounts from the measurements. Periods of high stray light levels have been removed from the record. An improvement to the optical configuration to avoid scattered light interference is described in section 4.3 under future directions.

## 2.4 Path averaged temperature measurement

Significant differences of up to 5°C became apparent between measurements of temperature from the point sensors located at the instrument and at the weather stations at each end of the optical path. An effective path-averaged temperature for each measurement is preferable to a point measurement, and was therefore retrieved from the spectra themselves by allowing temperature to be an adjustable parameter in the least-squares fit. The IUP station temperature was used as the initial estimate for the fit. Temperature was retrieved from the $CO_2$ window at 4980 cm$^{-1}$ (Figure 4b) which has good signal to noise ratio and absorption lines with a range of temperature sensitivities. Figure 5 illustrates typical temperatures and differences over a period of four sunny days – there is a systematic offset, with the point measurement always higher relative to the path average, and larger differences during daytime. This may be due to the thermal mass of the building on which the weather station was located or radiative heating of the sensor, while the open optical path is typically 10-30 m above the ground and buildings in free air. We expect the retrieved path averaged temperature to be a better estimate of the true path averaged temperature; this is confirmed when used to fit $O_2$ as described further below, as it led to less artefact diurnal variability in the retrieved $O_2$. The $CO_2$-spectrum-derived path-average temperatures were therefore used in all spectrum re-analyses in other spectral regions.

## 2.5 Instrument lineshape (ILS) characterisation

To check the instrument lineshape function (ILS) of the FTS, we followed Frey et al. (2015), by measuring the spectrum of water vapour in a short-path reference spectrum over a pathlength in air of approximately 3 m and fitting it using both MALT and Linefit (Hase et al., 1999) programs. Assuming the nominal field of view (FOV) of the FTS of 7.2 mrad, we found a linear drop in modulation efficiency to 0.67 at the maximum optical path difference. Alternatively, setting the modulation to its nominal value of 1.0 and fitting the field of view, we retrieved an effective FOV of 10.8 mrad. The effective ILS width is thus approximately 30% broader than the nominal value for a perfect optical system. This is consistent with the short focal length optics and aberrations in the compact optical system. The ILS is shown in Figure 6. The full width at half height is 0.58 cm$^{-1}$, equivalent to 0.12 nm at 7000 cm$^{-1}$ (1428 nm) and 0.24 nm at 5000 cm$^{-1}$ (2000 nm).

## 3 Results

All raw mole fractions (except water vapour) were converted to dry air mole fractions using the path-averaged water vapour amount retrieved from the same spectrum:

$$x_{dry} = \frac{x_{wet}}{1 - x_{H2O}}$$

### 3.1 Oxygen, $O_2$

Retrieval of the $O_2$ mole fraction from the 1.27 μm (7880 cm$^{-1}$) band (Figure 4a) provides a system check since the $O_2$ mole fraction is constant and well known, 0.2095 relative to dry air. Initial retrievals using the weather station pressure and temperature displayed diel variations of measured $O_2$ of the order of 1-2% that were reduced significantly using path-averaged

temperatures derived from the $CO_2$ spectrum fit, as described above. The $O_2$ measurements for the whole period are shown in Figure 7. The positive spikes observed regularly near 18:00-19:00 local time on clear sunny days are due to direct sunlight scattered into the FTS and detector as described in the previous section - when the solar beam path is from the west at low elevation and approximately aligned with the optical path (Figure 2), solar radiation is back-reflected from the retroreflectors and captured by the telescope. Corresponding spikes are also seen in $CO_2$ and $CH_4$ records and have been filtered to remove

all data where the raw retrieved $O_2$ mole fraction is greater than 0.225.

The mean mole fraction (excluding evening scattered sunlight anomalies) is 0.217, a bias of +3.6% (OP – in situ) from the known value of 0.2095. This is larger than the ~+2% bias found consistently at all TCCON sites, where it is attributed to inaccuracies in the spectroscopic line parameters (Wunch et al., 2010). Biases are discussed further in section 4.

### 3.2 Water vapour, $H_2O$ and HDO

Water vapour provides a further check of the FTS measurements against weather station humidity. (The in situ analyser does not measure ambient water vapour as the sample is dried for measurement.) $H_2O$ and its deuterated isotopologue HDO were co-fitted in a window 4910 - 5080 cm$^{-1}$ (Figure 4b, Table 1) and results are shown in Figure 8. δD was calculated as

$$\delta D = \left( \frac{(HDO/H_2O)_{air}}{(HDO/H_2O)_{SMOW}} - 1 \right) * 1000‰$$

where $(HDO/H_2O)_{air}$ is the measured isotopologue ratio and $(HDO/H_2O)_{SMOW}$ is the corresponding reference ratio for Standard Mean Ocean Water. The spectroscopically measured water vapour amount is in excellent agreement with the weather station record, with a 6% high bias which may be due in part to the humidity sensor itself. The uncalibrated mean δD is -68 ± 59 ‰, somewhat higher than recent summer measurements near Zurich, 230 km south of Heidelberg, -120 to -180 ‰

(Aemisegger et al., 2012). However the precision of the δD measurements is not sufficient to distinguish any variability related to temperature, and we do not analyse the δD results further here.

## 3.3 Carbon dioxide, $CO_2$

As is the case for $O_2$, the raw OP $CO_2$ mole fractions retrieved from the spectra are systematically higher than the calibrated in situ measurements at the IUP end of the open path. We attribute this bias to a calibration scale difference between the SI-traceable WMO scale of the in situ measurements and the uncalibrated OP measurements which are derived from spectrum fitting based on Hitran line parameter data and a spectrum model. To estimate the bias, we take the mean ratio of the OP to calibrated in situ measurements at wind speeds above 6 m s$^{-1}$ when the atmosphere is most likely to be well mixed and real differences between point and open path measurements are minimal. The bias is +2.5% (~10 ppm) and all raw OP data have been scaled down by a factor of 1.025 in the following discussion.

The bias-corrected OP and calibrated in situ measurements are shown in Figure 9, together with their differences. Figure 10 shows the differences plotted (a) against wind speed, (b) against wind direction, and (c) as a histogram. The data are discussed in section 4.

## 3.4 Methane, $CH_4$

Similar analyses for $CH_4$ are shown in Figure 11 and Figure 12. The mean OP - in situ difference for windspeeds >6 m s$^{-1}$ is +3.0% (~60 ppb). In this case there is a significant positive tail in the distribution of differences at all windspeeds (Figure 12) which increases the mean bias  for windspeed > 6 m s$^{-1}$; for the bulk of the data with windspeed < 2 m s$^{-1}$, the bias is 0.7% (17 ppb).

## 3.5 Carbon monoxide, CO

Absorption by the UV-quartz retroreflectors below 4600 cm$^{-1}$ in the region of the CO overtone band centred near 4300 cm$^{-1}$ prevents analysis of CO from these spectra. With more appropriate IR quartz, glass or hollow mirror retroreflectors of higher transmission in this region, a simulation of the resultant expected spectra based on the performance achieved with the current system suggests CO measurements with a 5-minute measurement averaging time would provide repeatability of the order of 5-10 ppb, which would be sufficient precision to resolve real variability in polluted urban environments.

## 3.6 Nitrous oxide, $N_2O$

$N_2O$ absorbs only weakly in the NIR. Analysis of the spectra in the strongest available band centred at 4730 cm$^{-1}$ provides a mean and standard deviation of the $N_2O$ mole fraction over the whole measurement period of $353 \pm 680$ ppb.  While the mean is realistic, the precision is not sufficient to detect meaningful changes in $N_2O$ amounts, which are small (a few ppb) due to

the weak sources and sinks and long lifetime of $N_2O$. A stronger band near 4415 cm$^{-1}$ would become accessible with glass retroreflectors, but would provide only a factor of two improvement.

## 4 Discussion

### 4.1 Precision, accuracy and open path – in situ bias

*Precision of measurements*

Table 2 and Figure 13 show Allan deviations (AD, the square root of Allan Variance (Werle et al., 1993)) for open path and in situ $CO_2$, $CH_4$ and $O_2$ measurements and the open path – in situ differences. The ADs in Table 2 and Figure 13 were calculated from the period 11 Aug 06:00 - 27 Aug 18:00 when diurnal variation was minimal and short term repeatability can be best estimated; they are presented for 5 min (single measurements), 1 hour and 6 hour averaging times. The 5 minute ADs

for the raw data provide upper limits for the instrument or measurement noise, since the variability over 5 min is dominated by instrument noise but there is also the possibility of a small contribution from atmospheric variability over 5 min time scales. For comparison, a smoothed curve through the raw data was subtracted from the raw data to remove the gross atmospheric variation (2$^{nd}$ order Savitzky-Golay smoothing, 15 points, approx 1-hour smoothing) and ADs recalculated (hereafter called "smoothed-subtracted" data). Five-minute ADs and the standard deviations of the smoothed-subtracted data are similar to the

ADs of the raw data at 5 min and are also shown in Table 2 and Figure 13; the smoothed-subtracted ADs decrease with averaging time out to 6 hours approximately as expected for random noise. The 5 min Allan deviation values are ~1.7 ppm (0.4%) for $CO_2$, 23 ppb (1.2%) for $CH_4$ and 0.0016 (0.7%) for $O_2$. For in situ measurements they are lower, reflecting the better repeatability of the in situ analyser: 0.63 ppm (0.15%) for $CO_2$ and 2.1 ppb (0.1%) for $CH_4$. We take these values as our best estimates of the 1-$\sigma$ repeatability of the measurements due to the instrument noise with minimum influence from

atmospheric variability.

For both open path and in situ $CO_2$ the AD increases with averaging time to ~ 9-11 ppm at 6 hours, reflecting the increased atmospheric (mostly diurnal) variability over the longer time periods of 20-40 ppm peak-peak. For open path $CH_4$ the increase in AD with averaging time is not as pronounced for OP data (23 to 40 ppb) because the measurement noise and atmospheric

variability are of comparable magnitudes. Diurnal variability of $CH_4$ is not as pronounced as for $CO_2$. For in situ $CH_4$ data the AD increases from 2 to 13 ppb. For $O_2$ there is no natural variability and the AD decreases with averaging time (0.0016 to 0.001 mole fraction).

For $CO_2$ the 5 minute AD of OP - in situ differences is also 1.7 ppm but remains approximately constant over averaging times

30 up to 6 hours, reflecting real (non-random) OP - in situ differences over hourly timescales. Over the full dataset 10 Jul – 4 Nov that includes periods of greater atmospheric variability, the 6-hour AD increases to 3.0 ppm. The distribution of the differences across all data appears near-normal with standard deviation 6.3ppm (Figure 10 (c)), but over shorter timescales can be seen

not to be simply random (Figure 9). These ADs and standard deviations taken together reflect that the actual variations of OP - in situ differences are 2-4 times larger than the 5 minute OP measurement noise of 1.7 ppm. For such a normal distribution of differences with standard deviation 6.3 ppm and a 1-$\sigma$ measurement repeatability of 1.7 ppm, approximately 40% of the measured differences lie more than two measurement standard deviations from the mean and may be considered atmospherically significant.

For $CH_4$ the 5 minute AD of OP - in situ differences increases slightly with time due to real atmospheric variability. The distribution of differences also appears near-normal over the whole dataset but with short term non-random variations and a positive tail due to two significant enhancements in OP $CH_4$ in August and September; the standard deviation of the distribution is 90 ppb (Figure 11, Figure 12); as for $CO_2$, 2-4 times the 1-$\sigma$ measurement repeatability.

*Open path – in situ bias*

Raw OP measurements are biased high relative to WMO-calibrated in situ measurements at the IUP (western) end of the path, +2.5% for $CO_2$, +3% for $CH_4$ and +3.6% for $O_2$. Quantifying these biases relies on the assumption that the atmosphere is well mixed along the open path for windspeeds > 6 m s$^{-1}$ and that there are no actual mole fraction differences under these conditions. For comparison, TCCON measurements of total columns over much longer atmospheric paths (typically > 10km) have consistent network-wide biases of approximately -3% for $CO_2$, -4.4% for $CH_4$, and +2% for $O_2$. (The TCCON network wide bias for $O_2$ is derived from the comparison of retrieved column $O_2$ amount with atmospheric pressure). The network wide biases for $X_{CO2}$ (= column$CO_2$/column$O_2$*0.2095) and $X_{CH4}$ (= column$CH_4$/column$O_2$*0.2095), which include and partially cancel the biases in both target species and $O_2$, are -1.0 and -2.4% respectively relative to in situ measurements over the atmospheric column with WMO-scale calibrated analysers (Wunch et al., 2010, updated 2014).) The biases are also similar in magnitude to those seen in uncalibrated mid IR OP and in situ FTIR studies (Smith et al., 2011; Griffith et al., 2012). Thus the observed biases in this study are generally consistent in magnitude with though not the same as other comparisons of FT spectroscopy with WMO-calibrated in situ measurements. As shown in the next paragraph, they are also consistent with an assessment of systematic errors in the retrievals of path-averaged mole fractions from open path infrared spectra.

Table 3 presents the sensitivity of mole fraction retrievals from the spectra to realistic uncertainties in input parameters and choices in the retrieval. Details are given in the caption to Table 3. There is no dominant single source of uncertainty; the main contributors are derived from uncertainties in spectroscopic data, the instrument lineshape, stray radiation, and details of the fitted spectral window. A simple quadrature sum of the estimated systematic errors (4.5% for $CO_2$, 3.3% for $CH_4$ and 5.9% for $O_2$) is larger than the observed systematic biases relative to calibrated in situ measurements; thus the observed biases are consistent with our a priori estimates of systematic errors. Although the open path measurements in this work and TCCON measurements use the same general spectral region, the near IR, there is no reason to expect that the biases would be the same

in both cases. The measurements differ in spectral bands analysed, spectral resolution and instrumentation, and most input parameters listed in Table 3.

Data from recent work using broadband DOAS and laser-based long open path techniques are shown for comparison in Table
4. Compared to conventional DOAS with a grating monochromator, array detector and the same long path fibre-telescope optics (Sommer, 2012; Saito et al., 2015; Somekawa et al., 2011), the FTS system achieves greatly improved repeatability. Compared to more recent work with dual frequency comb laser spectroscopy (Rieker et al., 2014; Waxman et al., 2017), the repeatability is less by about a factor of two. The frequency comb was operated over a longer pathlength with shorter measurement times and achieved lower bias when compared to co-located in situ measurements, but at this stage of
10 development is less portable for remote field measurements and applicable only to a narrower range of species. The FTS setup has advantages in terms of mobility and costs.

## 4.2 Comparison of open path and in situ measurements

From the preceding discussion, measured differences between open path and in situ measurements are only ~2-4 times the OP measurement repeatability. Actual differences are thus not well distinguished from measurement noise, and difficult to assign
unequivocally to specific sources. The discussion of differences is therefore brief.

$CO_2$

For bias-corrected $CO_2$ there is a mean OP – in situ difference of -3.2 ppm (in situ > OP) at low windspeeds relative to assumed well mixed conditions at wind speeds > 6 m s$^{-1}$. This difference is larger at night (-4.5 ppm) than during the day (-2.0 ppm), with a slight tendency to be larger for winds from the SE. This diurnal dependence of $CO_2$ difference could in principle be
partly due to time-of-day-dependent changes or errors in systematic temperature measurement (see section 2.4 and Figure 5), but in practice there is no correlation between OP – in situ $CO_2$ difference and the difference between weather station and path-averaged temperature ($R^2$=0.0003, 0.1 ppm °C$^{-1}$). The corresponding local source of $CO_2$ leading to higher $CO_2$ amounts at the IUP end of the path is unlikely to be local traffic from the nearby main road, Berliner Strasse, with most traffic and activity during daytime. The more distant Heidelberg city centre is distributed along the south bank of the river, and would be expected
to affect both OP and in situ measurements more equally. The most likely $CO_2$ contribution which is higher at night but lower during the day is the biosphere, with respiration at night and photosynthetic drawdown during the day, but it is not immediately clear why this would be more prevalent in the in situ measurements than the open path since trees and plants in the local area are quite evenly distributed. Agricultural areas to the NW may play some role. To summarise, we find that the measured differences are probably significant at a level of a few ppm, but not sufficiently clear above the measurement noise to be able
to draw any definitive conclusions or to assign to any specific sources or sinks.

CH4

For bias-corrected $CH_4$, there is also a mean negative difference in situ – OP difference at low windspeeds relative to windspeed > 6 m s$^{-1}$ (-44 ppb) which is also larger at night (-53 ppb) than during the day (-32 ppb). There is no dominant wind direction for these negative differences, and as for $CO_2$ the source is unclear. For $CH_4$ there are two distinct episodes of positive differences where OP measurements are significantly higher than in situ, around 9 August and 5 September. The August period corresponds to winds from NW of the IUP, while for the September period the enhancements are broadly distributed from the eastern sector. In both periods the enhanced differences occur mainly at night. The observed differences in OP $CH_4$ relative to in situ measurements are only marginally greater than the OP measurement stability and repeatability and are difficult to quantify or assign with any certainty to specific atmospheric conditions or local sources or sinks. $CH_4$ relative precision is lower than for $CO_2$ because of both the absolute strength of the $CH_4$ absorption features and their strength relative to overlapping water vapour absorption (Figure 4). There is no correlation between the OP-in situ differences and coincident water vapour amounts derived from the same spectra, suggesting that the $CH_4$ differences are not an artefact due to spectra overlap. There are numerous possible small, local point sources, such as natural gas or wastewater piping leaks, that may affect the observed differences, but with this level of precision, detailed interpretation can only be speculative.

The OP-in situ differences and geographical scales of these measurements approach the accuracy and resolution of developing regional scale models such as the Weather Research and Forecasting model (WRF) in high resolution mode (Viatte et al., 2017). A detailed high resolution modelling analysis of the measurements presented here might help in interpreting the observed in situ – OP differences, but is beyond the scope of this paper.

**4.3 Future improvements**

This study was made with available instrumentation in a restricted timeframe as a pilot study of the open path FTS technique in the NIR and did not optimise some aspects of the measurements. Several options are available to improve the accuracy and precision of the OP-FTS-NIR measurements:

- Interferences from stray radiation: especially at low solar elevations, background (stray) radiation is modulated and detected by the interferometer and leads to broad enhancements and spikes in measured concentrations. This can be almost entirely removed by reversing the source and detector in the optical system shown in Figure 1, first modulating the source in the interferometer before transmission over the open path. With this option stray environmental radiation such as direct or scattered sunlight is viewed directly by the detector and not modulated by the interferometer; it does not contribute to the Fourier-transformed infrared spectrum. This option was not possible with the available optics and spectrometer for this pilot study, but will be incorporated in the next build. With the present system, increasing the frequency of the background stray light measurements (1 per hour in this work) would allow better correction for stray light interferences due to short term variations in stray radiation, but at the cost of lower precision, measurement time and duty cycle.

- Increased optical throughput: using a brighter source and/or larger telescope and retroreflector area will translate directly into lower measurement noise and improved repeatability. This is particularly true of retroreflectors, which had a total area of around 510 $cm^2$ compared to the telescope primary mirror area of 700 $cm^2$. A close packed retroreflector array large enough to capture the (slightly divergent) open path beam could thus improve precision by a factor of about two for the same primary telescope aperture.

- Extension to include CO: for urban studies the measurement of CO is advantageous, both for its intrinsic interest and as a tracer for combustion sources of other trace gases. In this work we used available UV quartz retroreflectors optimised for UV/vis DOAS measurements. The transmission of UV quartz cuts off below 4500 $cm^{-1}$, precluding CO measurement in the overtone band around 4300 $cm^{-1}$. The use of corner cube retroreflectors with transmission to 4000 $cm^{-1}$ (for example hollow mirror, BK7 glass or IR quartz) will allow measurements to extend to CO. A simulation with the measurement noise realised in this work suggest a CO measurement repeatability of a few ppb, which should be sufficient for studies in urban areas.

## 4.4 Conclusions and final comments

We have introduced a long open path Fourier Transform spectrometer operating in the near infrared over a 3.1 km return path in open air. The system is able to make measurements of several species simultaneously by virtue of the broadband nature of the spectroscopy. We have demonstrated measurements of $CO_2$, $CH_4$, $O_2$, $H_2O$ and HDO; with a minor variation of optics CO is also possible, which would be of advantage in urban environments. The spectrometer is reasonably portable, able to be tripod mounted, and requires power (~ 150 W) and shelter at only one end of the path, with a passive retroreflector array at the far end of the path.

We observe significant differences of the order of a few ppm for $CO_2$ and a few tens of ppb for $CH_4$ between the open path and point measurements 2-4 times the measurement repeatability. In the context of fine scale atmospheric models, which now provide kilometre scale resolution, open path measurements have the potential to bridge the gap between high accuracy point measurements and spatially-averaging atmospheric models. With improvements in precision and accuracy to be expected in both broadband (FTS) and laser based techniques, open path spectroscopy provides a valuable new tool for urban and regional scale studies.

## 5 Acknowledgements

This work was carried out as a sabbatical leave project by DG at the Institute for Environmental Physics, University of Heidelberg. DG thanks Ingeborg Levin, Ulrich Platt and members of the DOAS and carbon cycle groups for their contributions and collaboration in providing the laboratory and long path optical systems for the study. Geoff Toon, JPL, provided updated 2015 versions of GFIT line parameters .

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

**Tables**

Table 1. Details of spectral windows used for fitting. * In $O_2$ there is also a weak contribution from collision-induced continuum absorption which is fitted with the overall continuum.

| Species fitted | Interfering species co-fitted | Spectral region $cm^{-1}$ | Spectral region $\mu m$ |
|---|---|---|---|
| $O_2$ | $H_2O$ * | 7790 – 7960 | 1.26 – 1.28 |
| $CO_2$ | $H_2O$ | 4800 - 5050 | 1.98 – 2.08 |
| $CH_4$ | $H_2O$ | 5885 - 6150 | 1.63 – 1.70 |
| $H_2O$, HDO | $CO_2$ | 4910 – 5080 | 1.97 – 2.04 |

Table 2. Allan deviations for open path and in situ measurements and their differences.  The Allan deviation analysis is taken over the period 11 Aug 06:00 - 27 Aug 18:00 when diurnal variations were least.  *In the "smooth-subtracted" and standard deviation rows a smoothed curve through all the data has been subtracted from the raw data to remove the gross atmospheric variability and approximates the measurement noise. See text for further details.

| Allan Deviation | | $CO_2$ / ppm | | | $CH_4$ / ppb | | | $O_2$ |
|---|---|---|---|---|---|---|---|---|
| | | OP | in situ | diff | OP | in situ | diff | OP |
| **Raw data** | 5 min | 1.7 | 0.63 | 1.7 | 23 | 2.1 | 23 | 0.0016 |
| | 1 hr | 3.0 | 3.4 | 1.5 | 22 | 7.0 | 23 | 0.00092 |
| | 6 hr | 9.2 | 11 | 1.9 | 30 | 13 | 40 | 0.00097 |
| **\*Smooth subtracted data** | 5 min | 1.7 | 0.54 | | 22 | 1.8 | | |
| | 1 hr | 0.21 | 0.15 | | 2.5 | 0.51 | | |
| | 6 hr | 0.032 | 0.038 | | 0.59 | 0.068 | | |
| **\*Std.dev.** | | 1.6 | 0.82 | | 21 | 2.3 | | |

**Table 3. Sensitivity of retrieved mole fractions to retrieval inputs in the OP-FTIR measurements. Each input parameter or choice was varied by an estimate of its uncertainty in the MALT spectrum analysis and its effect on retrieved mole fractions calculated.**

| Quantity | Δ | Δ% | $CO_2$ | | $CH_4$ | | $O_2$ | |
|---|---|---|---|---|---|---|---|---|
| | | | ppm | % | ppb | % | molfrac | % |
| [1]Temperature/°C | +3 | +1% | +6.04 | 1.4% | +24.7 | 1.3% | +0.0032 | 1.5% |
| [1]Pressure/mb | +1.0 | +0.1% | -0.69 | 0.2% | -2.0 | 0.1% | -0.0002 | 0.1% |
| [2]Pathlength/m | + 3 | +0.1% | -0.4 | 0.1% | -1.8 | 0.1% | -0.0002 | 0.1% |
| [3]Linestrengths | | +2% | -8.0 | 2% | -36 | 2% | -0.002 | 2% |
| [3]Linewidths | | +2% | -4.88 | 1.2% | +5.4 | 0.3% | -0.0004 | 0.2% |
| [4]Zero offset | +0.01 | +1% | +10.5 | 2.5% | +21.0 | 1.1% | 0.0029 | 1.4% |
| [5]ILS width | | +5% | +5.0 | 1.2% | | | 0.0079 | 3.8% |
| [6]Window selection | | | | 2% | | 2% | | 2% |
| [6]Continuum polynomial | | | | 1% | | 0.1% | | 3% |
| [7]Fibre Residual | | | | <<1% | | <<1% | | <<1% |
| **Quadrature sum** | | | | **4.5%** | | **3.3%** | | **5.9%** |

**[1] Temperature and pressure errors affect retrieved mole fractions in two ways – proportionally through the dilution of air to calculate mole fraction from concentrations, and through the temperature and pressure sensitivities of linestrengths and lineshapes. From the net sensitivities, it can be seen that the errors are dominated by the dilution effects.**

**[2] Pathlength error propagates proportionally into the path average mole fraction, since the spectrum analysis retrieves the concentration- pathlength product.**

**[3] We estimate for a 2% error on Hitran linestrengths and linewidths – these errors are not well characterised (Toth et al., 2008).**

**[4] Adding a zero offset of 1% to the spectrum simulates the effect of 1% stray sunlight added to the spectrum, and can be taken as an estimate of the maximum effect due to stray light.**

**[5] The Instrument Line Shape (ILS) is fitted for every spectrum by allowing the FTIR field of view (FOV), phase error and frequency shift to vary in the least squares minimisation. The quoted error is calculated by forcing the width to increase by 5% above the best-fit value to estimate the effect of a non-ideal ILS.**

**[6] The selection of spectrum window to be fitted, and the number of terms in the polynomial used to fit the continuum, is somewhat subjective – the selections are based on visual assessment of the spectral residual and the minimum mean residual achieved. The sensitivity taken from the variation in retrieved concentrations across a range of "acceptable" window and baseline choices. Note the continuum choice for $O_2$ is more sensitive because the polynomial is effectively used to fit the unstructured pressure-induced continuum in the $O_2$ spectrum. Although we measured short path spectra every hour, in principle to characterise the continuum spectrum, using these spectra to define the continuum rather than fitting it did not improve fits, but added noise and an extra source of variability. All results were thus obtained with the continuum fitted with a 5-term polynomial.**

**7** The fibre optic coupling between telescope, source and detector introduces repeatable fringing and interferences in the measured spectra at about 1% of the measured signal intensity. These structures can be seen in the residual plots of Figure 4 and are quite reproducible over periods of days to weeks. They are larger than the underlying detector noise but much less than the trace gas absorptions, at least for $CO_2$ and $O_2$. Removing or co-fitting an average fibre residual spectrum during the fit makes only a small (<<1%) difference to the retrieved mole fractions because the fibre residual spectrum is itself derived from the least squares fits to real spectra and is approximately orthogonal to the target gas spectrum.

**Table 4. Comparison of repeatability and bias of long path techniques in the NIR region. [1] (Sommer, 2012; Saito et al., 2015; Somekawa et al., 2011); [2] (Rieker et al., 2014; Waxman et al., 2017)**

| | $CO_2$ / ppm | | | $CH_4$ / ppb | | |
| --- | --- | --- | --- | --- | --- | --- |
| | Open path | OP – in situ difference | | Open path | OP – in situ difference | |
| | Repeatability (Allan dev. 5 min) | Repeatability (Allan dev., 5 min) | Bias (uncorrected) | Repeatability (Allan dev. 5 min) | Repeatability (Allan dev., 5 min) | Bias (uncorrected) |
| **FTS (this work)** | 1.6 | 1.7 | 10 | 12 | 23 | 60 |
| **DOAS[1]** | 2-4 | - | - | >200 | - | >200 |
| **Freq. Comb[2]** | <1 | - | 3- 6 | <5 | - | 4-20 |

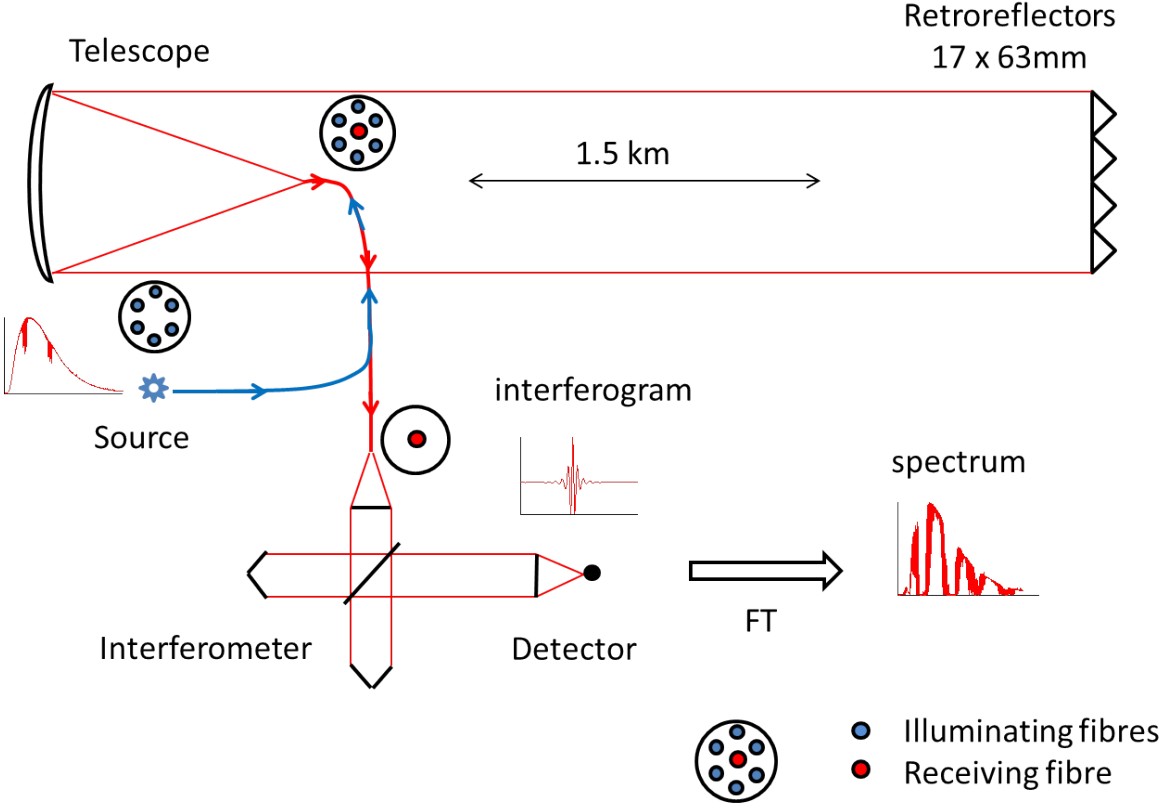

**Figure 1. Schematic drawing of the long open path FT spectrometer and optical system. Radiation from the source is fed close to the focus of the telescope through the outer bundle of six fibres (blue) and transmitted across the open path. The return beam is collected by the central fibre (red) and focussed onto the input aperture of the interferometer. The modulated beam from the interferometer is detected by the InGaAs detector and the resultant interferogram is Fourier transformed to provide the long open path spectrum.**

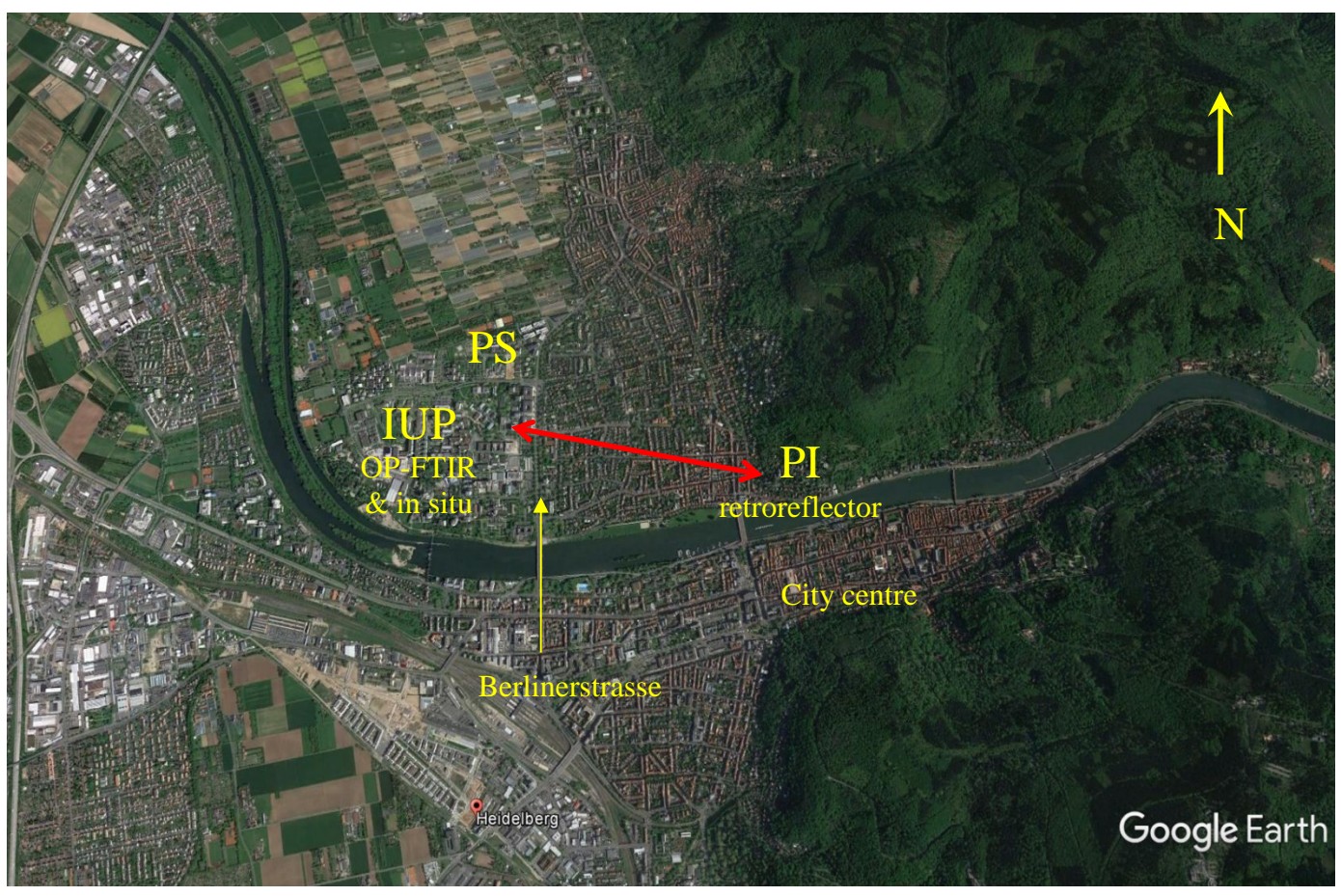

**Figure 2. Aerial view of Heidelberg and the 1.5 km measurement path. IUP = Institute of Physics (FTS and telescope, in situ measurements), PI = Physics Institute (retroreflector), PS = power station. The measurement path is mostly over residential areas. There is an extensive small-agricultural area to the N and NW**

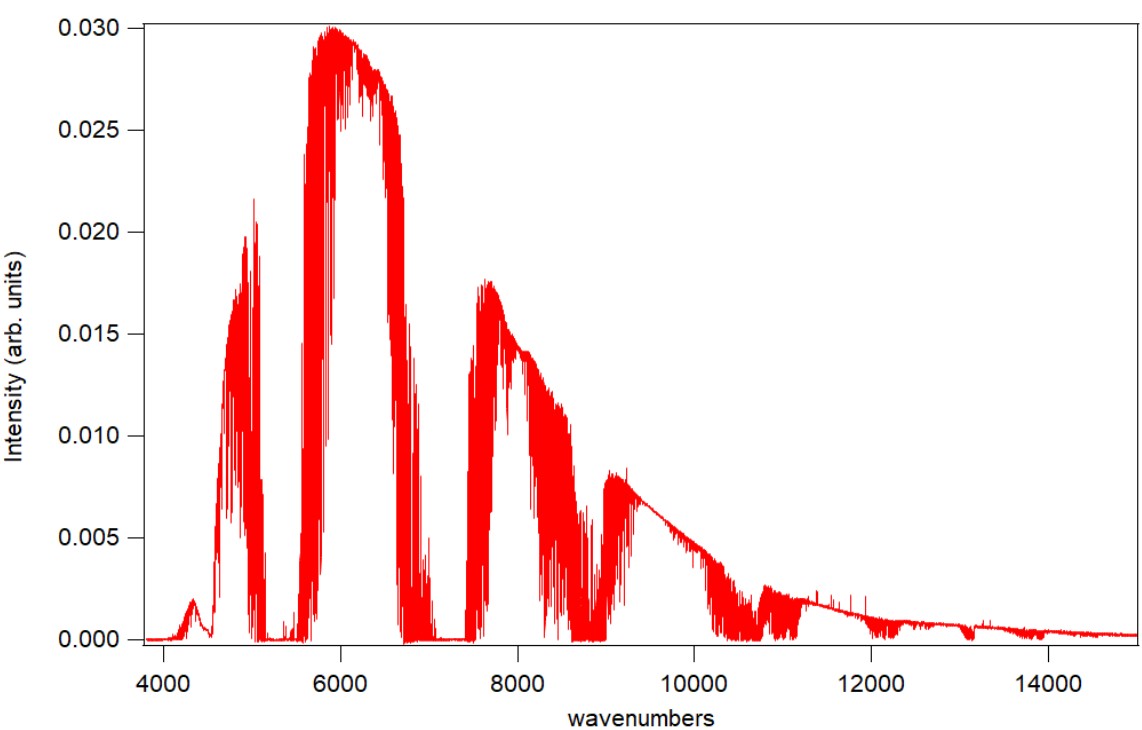

5    **Figure 3. Typical NIR long path spectrum, recorded 01 Oct 2014.**

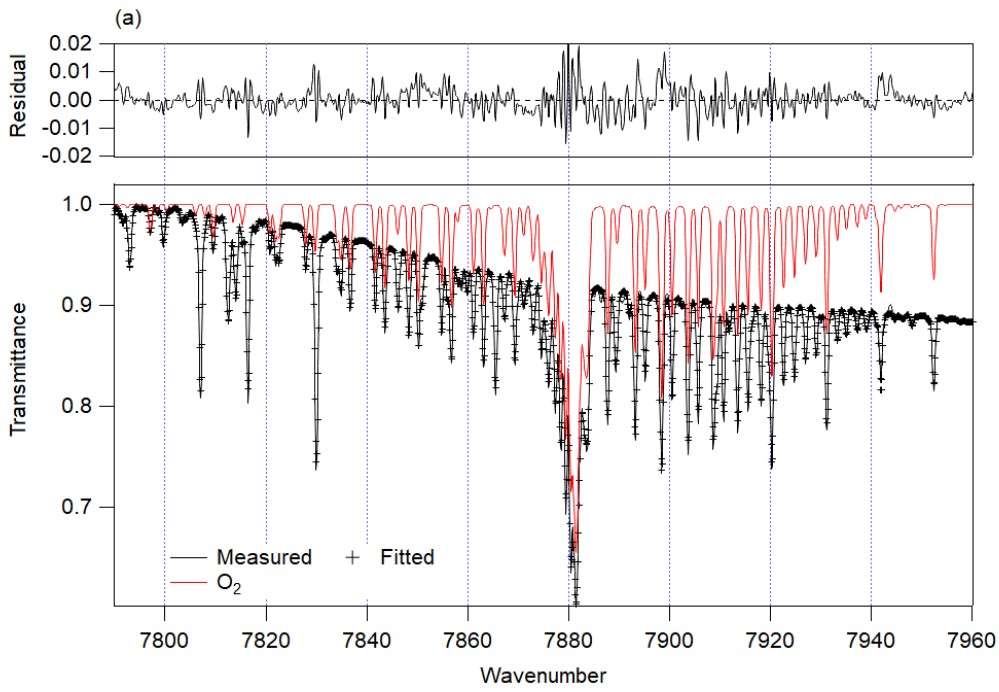

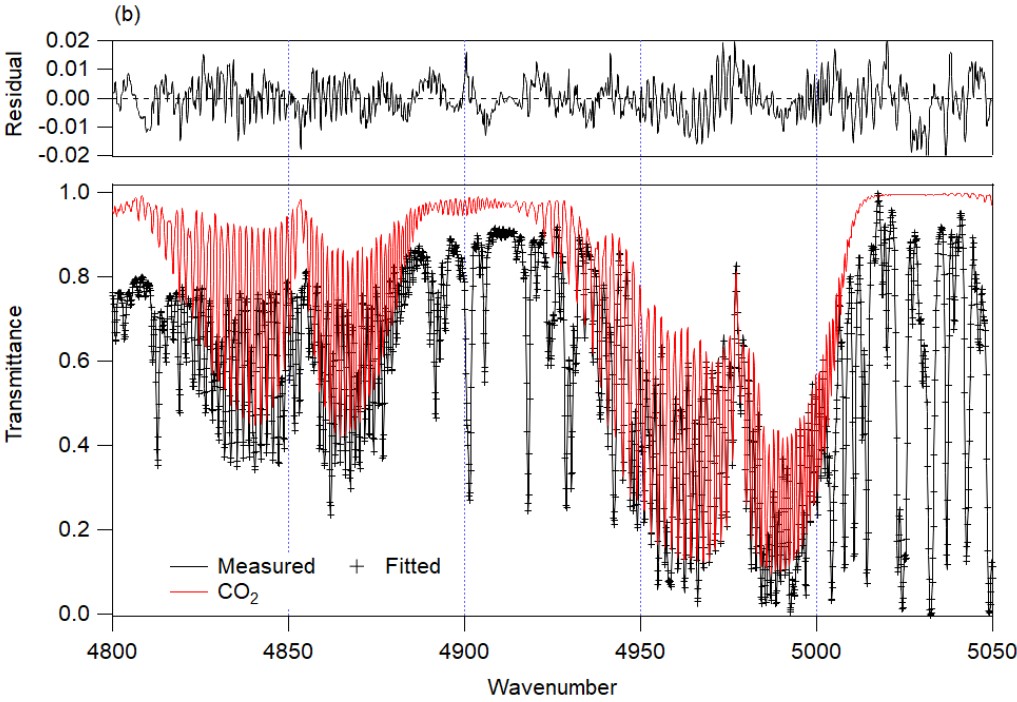

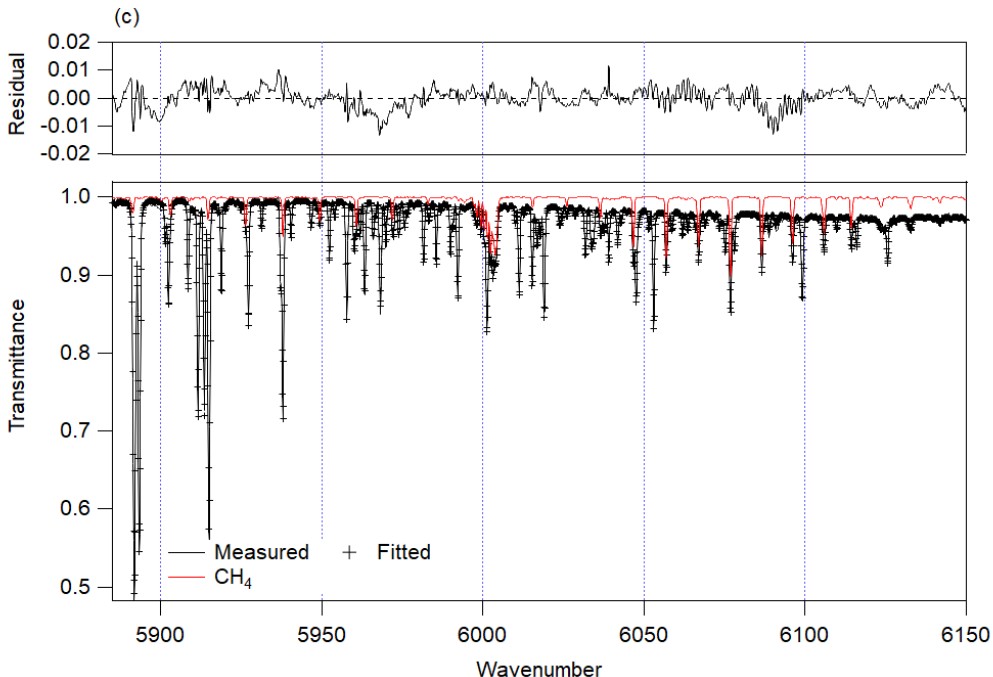

**Figure 4. Typical fits for (a) O₂ band centred near 7880cm⁻¹ (b) CO₂ bands centred near 4850 and 4980 cm⁻¹ and (c) CH₄ band centred near 6000 cm⁻¹. In each plot the target species is in red, and the remaining absorption is dominated by water vapour. See Table 1 for details af all interfering and co-fitted species.**

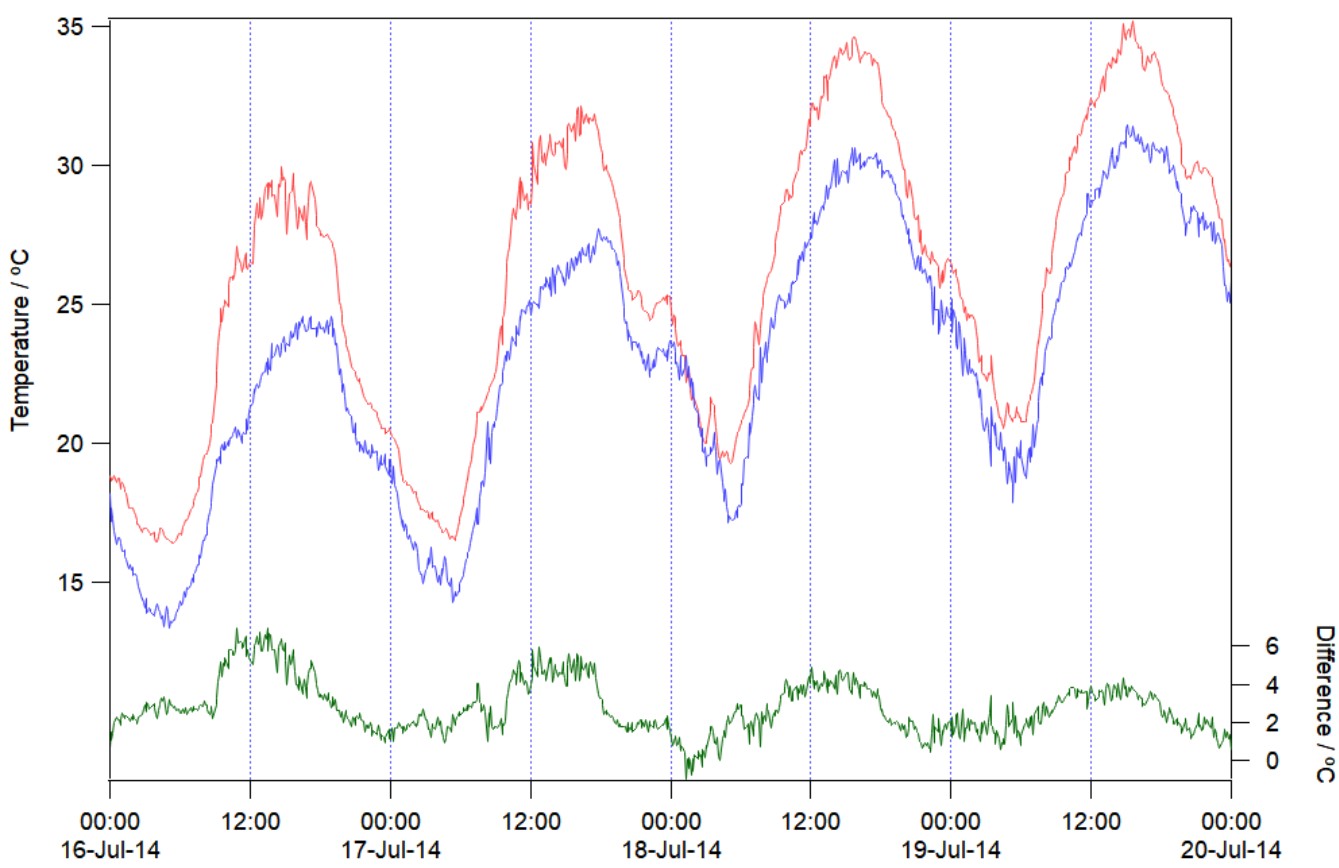

**Figure 5. Comparison of IUP meteorological station temperature (red) and spectrum-derived path averaged temperature (blue) for an illustrative period of 4 sunny days. The differences are plotted in green and range from 0 – 6 °C.**

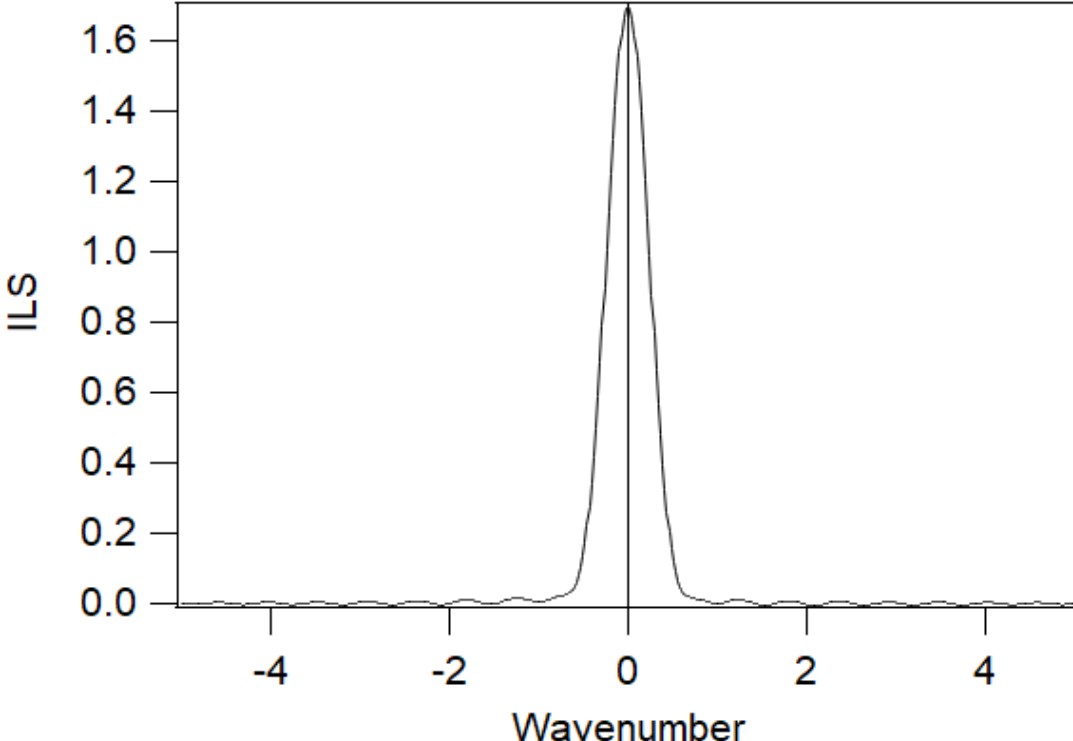

5    **Figure 6. Retrieved instrument lineshape function for the IRcube FTS at nominal 0.55 cm⁻¹ resolution. The measured half width at half height is 0.58 cm⁻¹.**

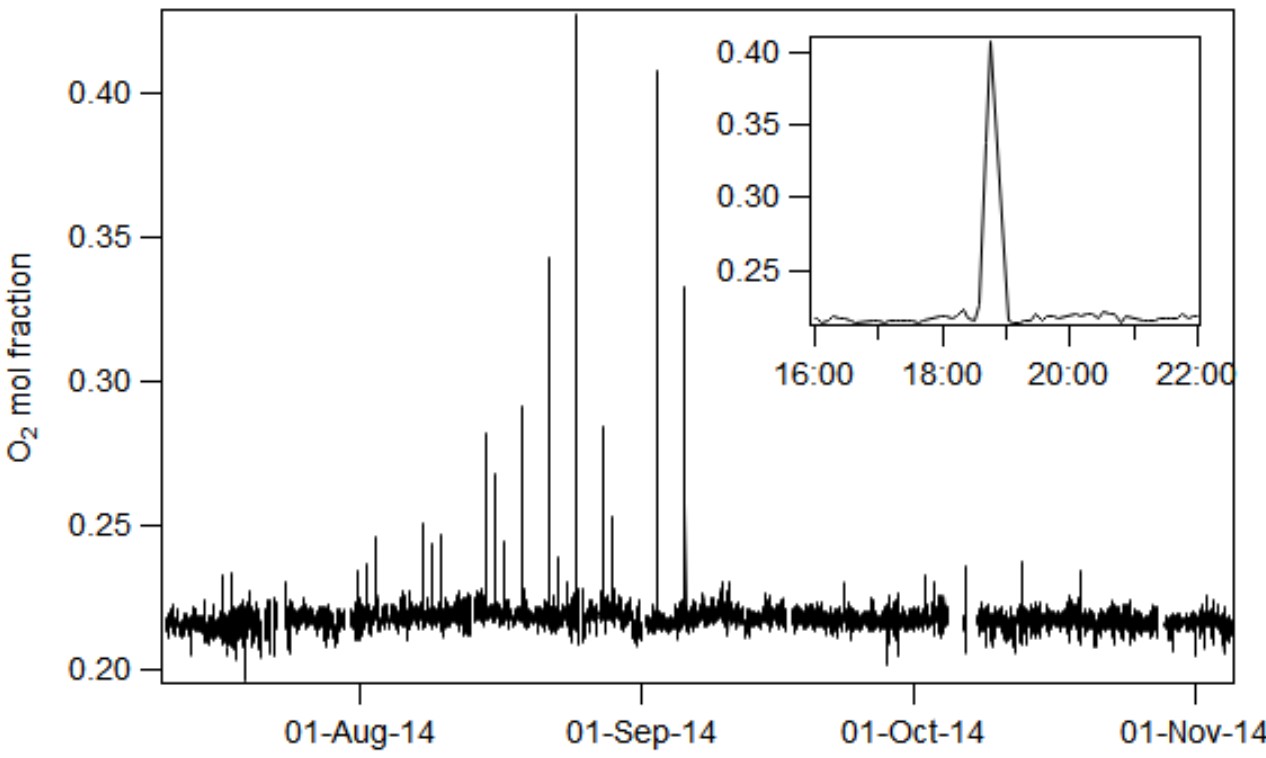

Figure 7. Measured $O_2$ mole fractions for the measurement period. The narrow spikes are artefacts due to stray solar radiation near 18:00 on sunny days, as discussed in the text. The insert shows details of the spike on 2 Sept 2014.

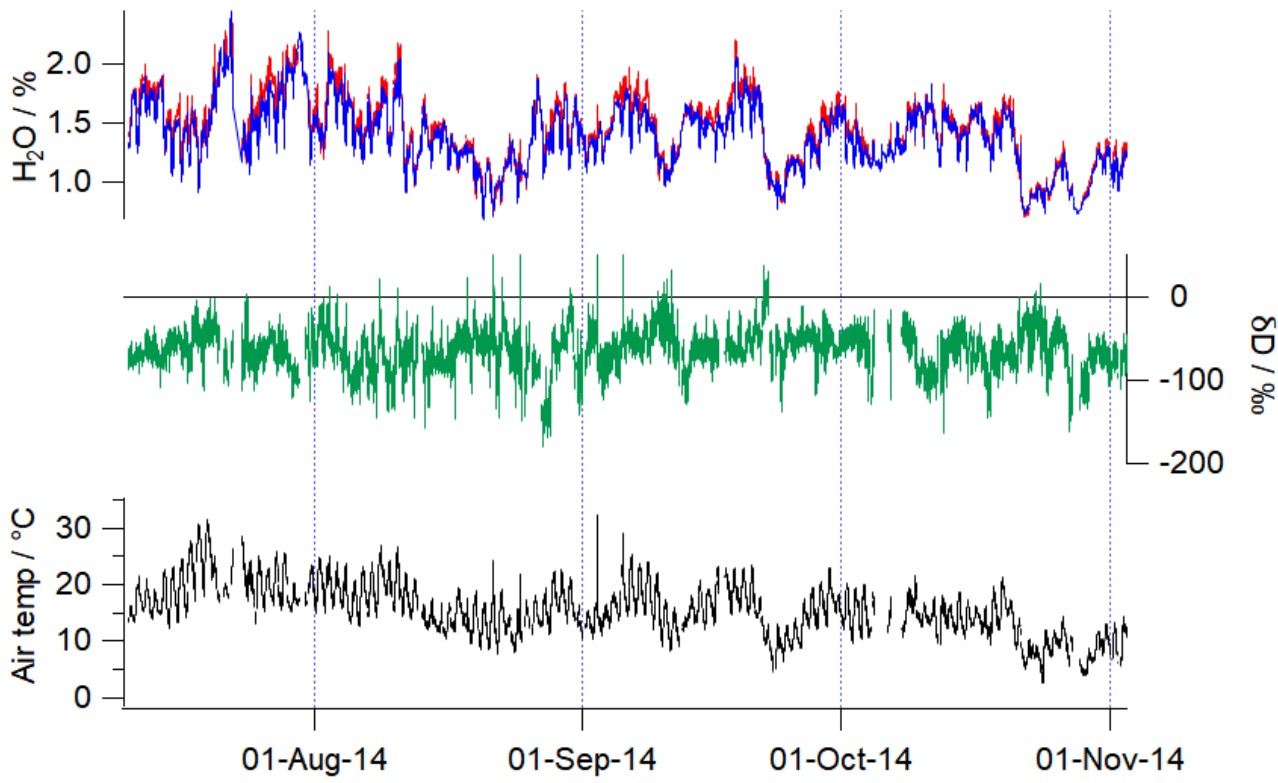

5 **Figure 8. Water vapour, δD and air temperature for the whole measurement period. In the upper panel the FTIR retrieved water vapour is in red and the IUP meteorological station absolute humidity in blue (as mole fractions in %).**

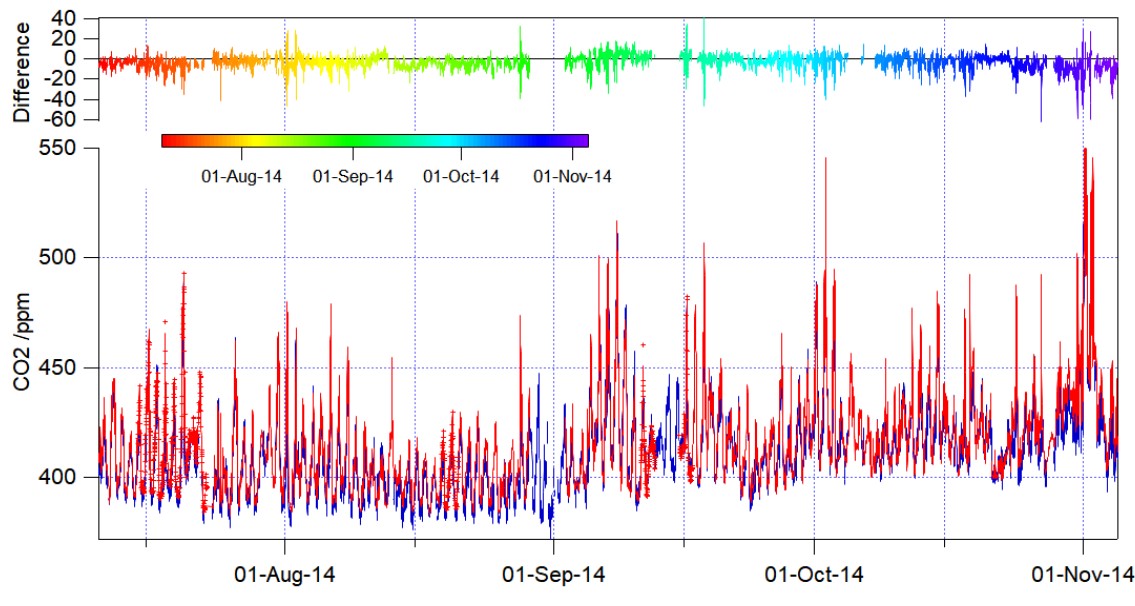

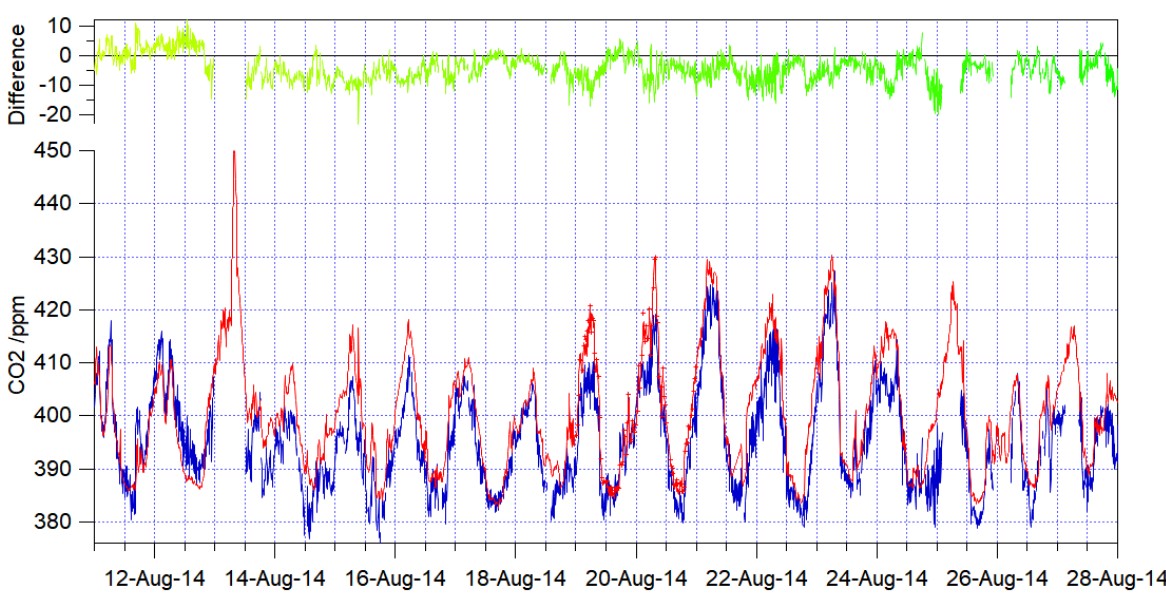

**Figure 9. Open path (blue), in situ (red) and difference (OP – in situ, coloured by time) measurements of CO₂. All raw OP data have been reduced by a factor of 1.025 (~ 10 ppm) to remove measurement bias relative to the in situ data. In the corrected data, there is zero bias for wind speeds > 6 m s-1 over the entire measurement period (see text for detail). (a) shows the whole measurement period. (b) illustrates a selected period with a consistent, real OP-in situ difference relative to the well mixed average.**

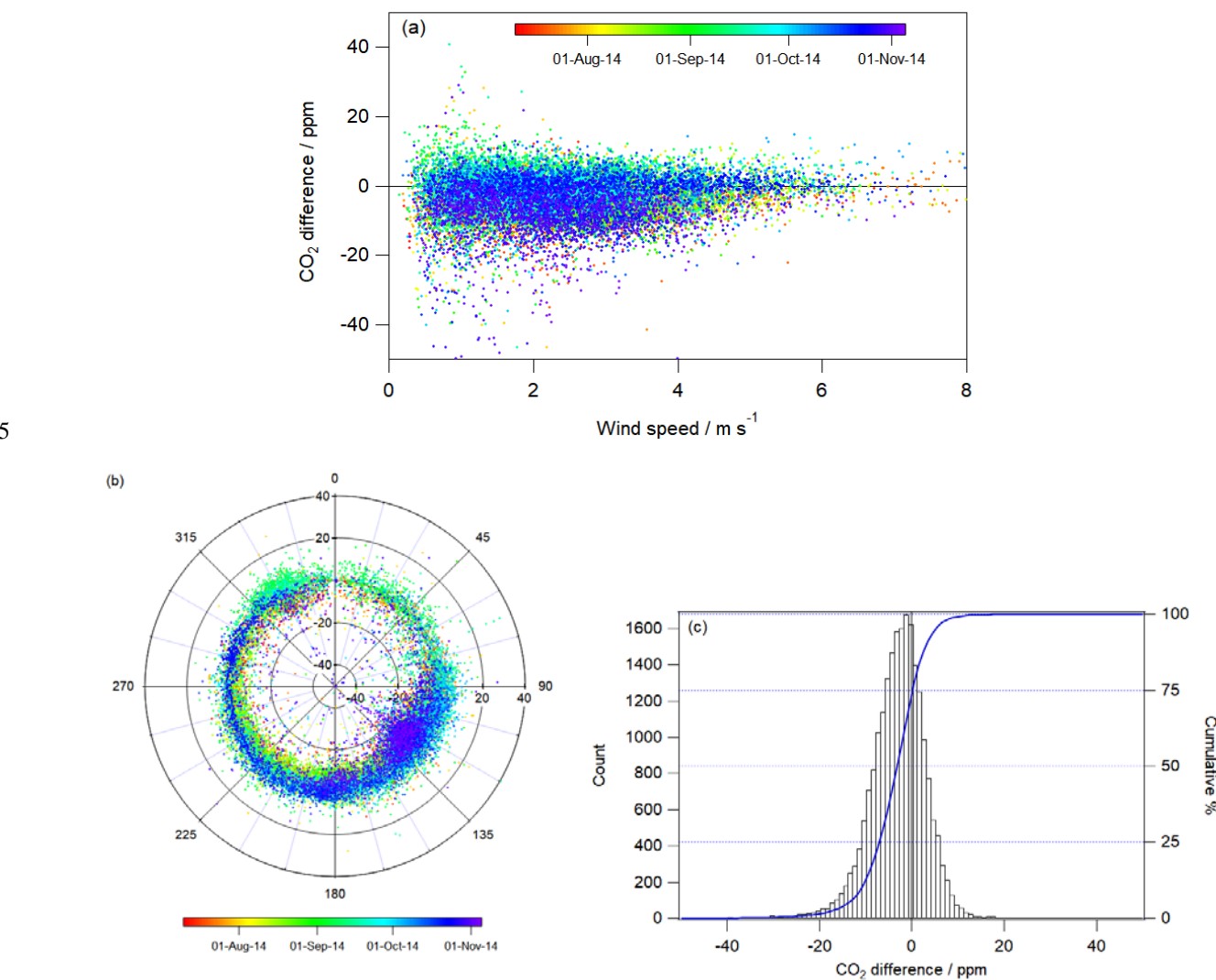

**Figure 10.** CO₂ mole fraction differences between open path and in situ measurements (OP – in situ) (a) vs wind speed (b) as a wind speed rose, and (c) as a histogram of the differences. The standard deviation of the distribution is 6.3 ppm. (a) and (b) are coloured by time to compare with Figure 9.

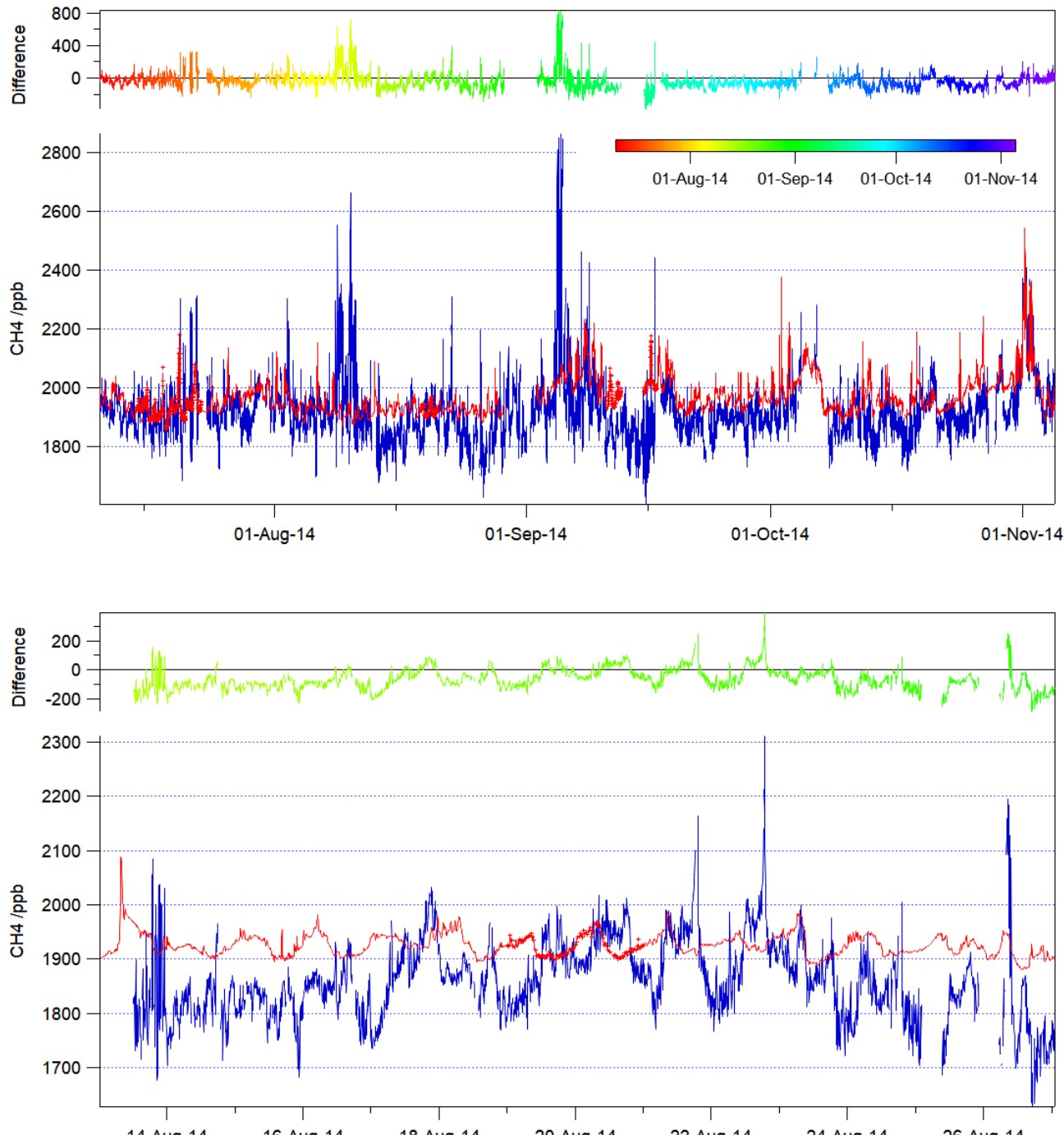

**Figure 11. Open path (blue), in situ (red) and difference (OP – in situ, coloured by time) measurements of CH4 for the whole measurement period. The uncalibrated OP data have been reduced by a factor of 1.030 (~60 ppb) to fit the in situ data for wind speeds > 6 m s$^{-1}$ (see text). (a) whole measurement period. (b) expanded period.**

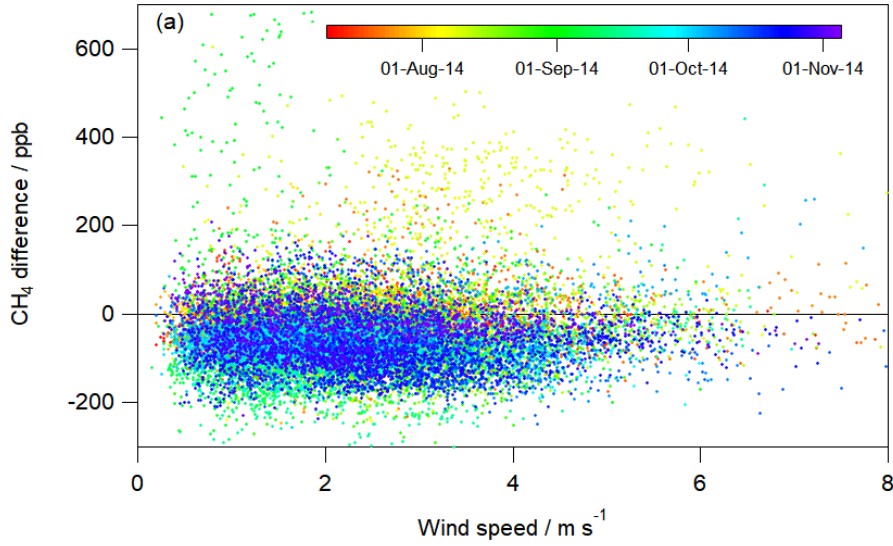

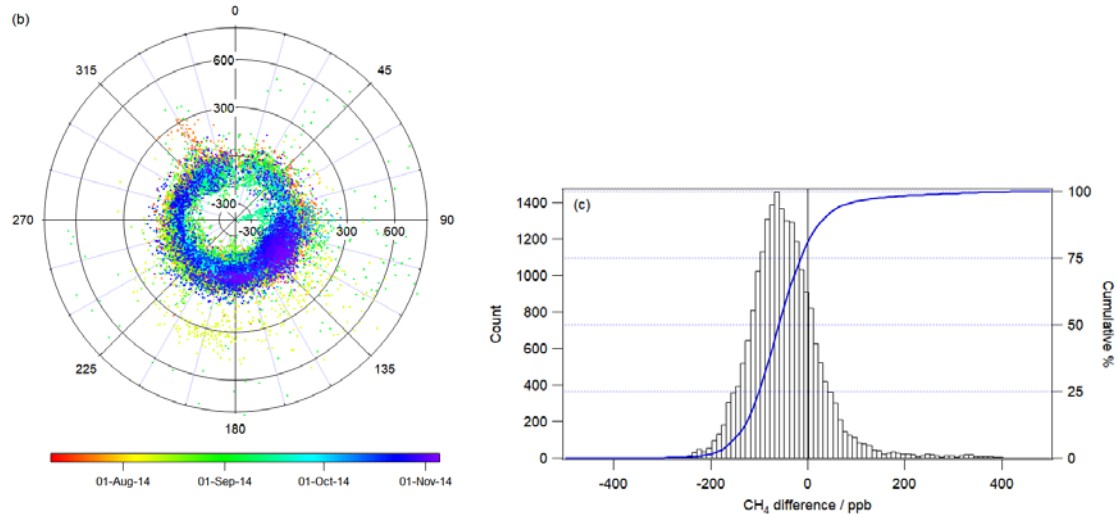

**Figure 12. CH₄ mole fraction differences between open path and in situ measurements (OP – in situ) (a) vs wind speed (b) as a wind speed rose, and (c) as a histogram of the differences. (a) and (b) are coloured by time to compare with Figure 11.**

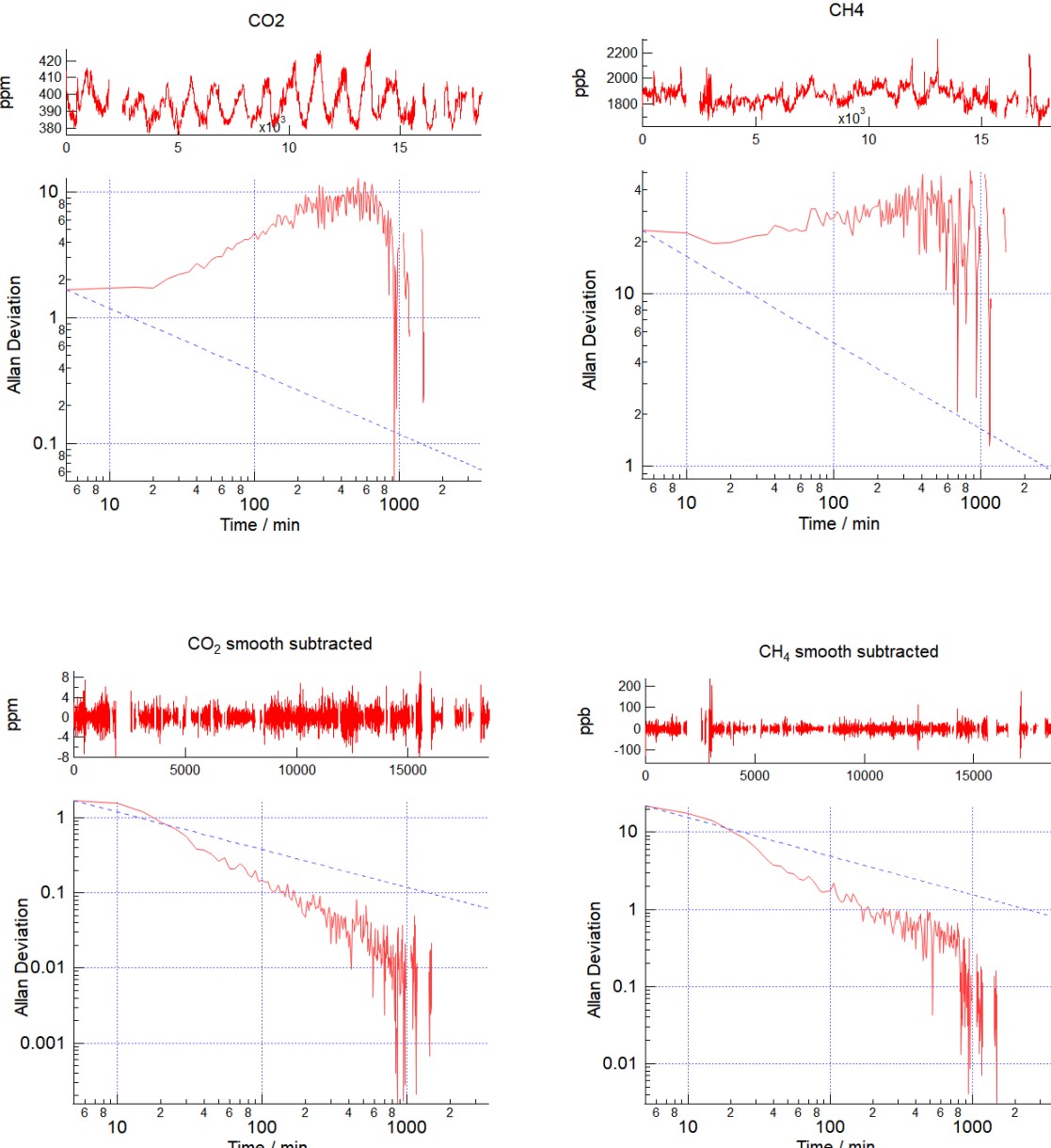

5    **Figure 13. Allan deviation plots for (a) OP CO₂ (b) OP CH₄ (c) smooth-subtracted OP CO₂ and (d) smooth-subtracted OP CH₄.  See text for details of subtraction of smoothed from raw time series.**