# Peer review of "Long open path measurements of greenhouse gases in air using near infrared Fourier transform spectroscopy"

_Atmospheric Measurement Techniques, 2017_

## Referee Comment (RC1) · Anonymous Referee #1 · 19 Sep 2017

In this work, Griffith et al. have deployed an open path near-IR FTS instrument to measure $CO_2$, $CH_4$, $O_2$, $H_2O$, and HDO over a 3 km round-trip path over the city of Heidelberg. These measurements were performed nearly continuously over the course of 4 months, and were compared against a WMO-calibrated in-situ point sensor instrument (also an FTS, though mid-IR) located at one end of the path. The authors are able to use an impressive 60% of the data that was collected over these four months. However, the authors find significant discrepancies between the measured $O_2$ concentrations and the known dry air mole fraction of $O_2$, as well as large differences between the $CO_2$ and $CH_4$ measured by the open-path FTS instrument and the point sensor FTS instrument. I have some comments that should be addressed prior to publication in AMT.

Major comments:

The authors collect multiple $CO_2$ bands in the spectrum shown in Figure 3, specifically a stronger band at 2.01 microns, a medium band at 2.06 microns, and a weaker band at 1.6 microns. The authors fit the first two bands, but not the one at 1.6 microns ($\sim6250$ cm$^{-1}$). Have they considered doing this and comparing the $CO_2$ retrieved between the two spectral regions?

The authors do not discuss the residuals observed in Figure 4. There are some fits at e.g. 7805 cm$^{-1}$ in the $O_2$ fit that look like imperfect Voigt fits, but the majority of residuals do not have this appearance. Do the authors have any ideas what causes these large residuals? Do the residuals change as a function of time of day, and could partially result from stray light?

The authors use $O_2$ simply as a "system check". Could they use it to filter out spectra that have been influenced significantly by stray light (which then results in the large spikes observed in the $O_2$ time series in Figure 7)? Do the authors have an explanation for the additional 1.6% $O_2$ bias beyond what is observed by TCCON? If the extra 1.6% discrepancy is due to e.g. unidentified instrumental systematics, could the authors use the $O_2$ to correct the $CO_2$ and $CH_4$ data for the same issues and possibly get better agreement between the open path and point source measurements? The ability to retrieve $O_2$ is one of this instrument's strengths compared to the frequency comb instrument and I think that the authors could capitalize on this strength more than they currently do.

Do the biases for $CO_2$ and $CH_4$ change if the authors use the same wind cutoff for "well mixed" conditions for both gases? They currently consider $CO_2$ to be well-mixed at wind speeds >6 m/s and $CH_4$ to be well-mixed at wind speeds >2 m/s.

I do not agree with the author's assessment that the additional difference between the in-situ and corrected open path $CO_2$ is the result of local emissions. Based on where the City Center is marked in Figure 3, emissions from the city center would affect primarily the open path measurements with S or SE winds as mentioned and would thus bias the open path instrument high relative to the point sensor, not low. Additionally, if there were very local traffic sources from Berliner Strasse, that should be evident as narrow, sharp spikes in the point sensor data (or at the very minimum, the two are likely to agree during most of the night when traffic is presumably minimal and there is no stray light) and that does not seem to be observed in the data

in Figure 9. Both time series have similar shapes, but with an offset between the corrected open path data and the in situ data that does seem to drift with time.

Do the authors have any idea what is causing the large discrepancy between the open path and in situ $CH_4$ data? Is the power station a natural gas station that would be expected to leak $CH_4$? Are there other local sources of $CH_4$ that are not geographically evenly distributed and might come from the WSW?

Minor comments:

Page 4, lines 13-14: the 3000 spectra that were thrown out due to poor visibility and other weather-related phenomena is approximately 11% of the of the total number of spectra. Why were the other ~30% rejected?

Page 4, line 25: When was the point sensor last calibrated against the WMO-GAW standards? How often does this instrument need to be calibrated? Could drift or lineshape change in this instrument cause some of the offset between the open-path and point sensor instruments?

Page 7, line 12: How are the $CO_2$ and $CH_4$ corrected to dry air mole fraction?

Page 9, lines 1-2: It is not clear whether the 2.5% and 0.7% biases observed for $CO_2$ and $CH_4$ include the correction or not. Is the total $CO_2$ bias 5%?

Figure 2: Please label the launch/receive and retroreflector ends of the path, as well as the location of the point sensor and Berliner Strasse. Please add a North indicator or compass rose. Is there a dominant wind direction that can be added to the plot as well?

Figure 7: A zoom in on one of the spikes would be nice so the reader can see that they are indeed sharp and centered at ~6 pm local time.

Figure 11: A zoom in on a time period similar to Figure 9 would be nice.

Figure 12: y axis in a and x-axis in histogram should be changed to ppb.

---

## Referee Comment (RC2) · Anonymous Referee #2 · 19 Sep 2017

Griffith et al present an open-path Fourier transform spectrometer operating in the near infrared with the ability to measure $CO_2$, $CH_4$, $O_2$, $H_2O$ and HDO. Such a system has the potential to be useful for small-scale, continuous greenhouse gas observations. The measurements were performed nearly continuously for 4 months over a 3-km-round-trip path, which is large enough to compare with high-resolution atmospheric models, and reached sensitivities of 1-2 ppm for $CO_2$ and 40 ppb for $CH_4$ with 5-minute measurement times. The authors attempt to quantify the bias and potential systematic errors by comparing to a WMO-calibrated point sensor located at one end of the open path. Overall, the manuscript provides a useful characterization of the system; however, there are some modifications and additional details that I think should be addressed prior to publication.

**Major Comments:**

First, for the precision of the measurements, could the authors please provide an Allan deviation from a well-mixed time period? I find this much easier to understand than just listing the precision in words. Also, why was one-quarter of the peak-to-peak used to estimate the precision instead of taking the standard deviation? Finally, the I found the terminology for variability, precision, uncertainty, and repeatability to be a bit confusing because multiple terms were used interchangeably throughout the manuscript (for example, uncertainty, precision, and repeatability). For clarity, could the authors use a consistent set of terms? I suggest perhaps variability for a time series of data and precision for the final derived result (i.e., remove uncertainty and repeatability).

Could the authors provide some more details about the spectral analysis, specifically: How were the wet measurements converted to dry mole fraction? What polynomial order was used in each spectral region? Was any other filtering or rejection done (e.g., were spectra rejected for anything other than poor signal) or were other corrections applied (e.g., "ghosting" corrections)? Was the ILS measured at multiple times to check for any drift in the ILS?

The authors note an addition 1.6% $O_2$ bias beyond what is observed in TCCON? Since the authors are using the TCCON spectral database, it does not seem that this added bias would be due to line parameters. Do the authors have a suspected source for this added bias? Is the bias constant in time?

Have the authors tried fitting $CO_2$ in the 1.6 μm band to compare to the 2 μm region?

For $CH_4$, do the authors have an explanation for what causes the positive tail in the distribution for >6 m/s wind speeds? This would imply that the open-path measurement reads higher than the point sensor, which seems odd at high wind speeds.

There is significant structure in the residuals shown in Figure 4 that is not discussed in Section 2.3. Note 7 of Table 2 briefly discusses some fringing in the residuals, which should be covered in Section 2.3. There also seems to be other structure in the residuals such as low-frequency baseline wobble and maybe also structure at the lines. Have the authors looked at the residuals

in more detail? How stable are the residuals? What causes the additional structures (they look to be larger than the residuals seen by TCCON)?

Finally, have the authors performed any laboratory measurements, e.g., using a multipass cell, with the system to characterize the bias in that configuration? This would be a useful check to see how the open path compares to a controlled system.

**Minor comments:**

In the introduction, there is no mention of OP-TDLAS or LIDAR systems (e.g., DIAL or IPDA). It would be worthwhile to mention these other technologies and how they compare.

Page 7, line 15: I was under the impression that the WMO scale is not directly SI traceable. Could the authors please clarify where the SI traceability comes from?

Page 10, line 4: CH4 needs subscript

Page 10, line 30: Typo in the units ($cm^{-2}$)

Section 4.3: Have you tried over a longer path?

Table 1: The values given for the $CO_2$ spectral range do not agree between the wavenumber and micron columns (i.e., 4800 $cm^{-1}$ = 2.08 um).

Table 2: in the $\delta O_2$ column, linestrengths row, what is the "()" for?

Fig 7: Perhaps zoom in vertically so that the majority of the time series can be seen more clearly. In addition, have the authors analyzed this data series for a diurnal cycle?

---

## Author Comment (AC1) · 12 Dec 2017

**AMT-2017-272_AuthorResponse_RC1**

Referee comments are in *italic*

*In this work, Griffith et al. have deployed an open path near-IR FTS instrument to measure CO2, CH4, O2, H2O, and HDO over a 3 km round-trip path over the city of Heidelberg. These measurements were performed nearly continuously over the course of 4 months, and were compared against a WMO-calibrated in-situ point sensor instrument (also an FTS, though mid-IR) located at one end of the path. The authors are able to use an impressive 60% of the data that was collected over these four months. However, the authors find significant discrepancies between the measured O2 concentrations and the known dry air mole fraction of O2, as well as large differences between the CO2 and CH4 measured by the open-path FTS instrument and the point sensor FTS instrument. I have some comments that should be addressed prior to publication in AMT.*

We thank both referees for their comments and the consequent improvements to the paper. Responses to general and specific comments are detailed below.

Major comments:

*The authors collect multiple CO2 bands in the spectrum shown in Figure 3, specifically a stronger band at 2.01 microns, a medium band at 2.06 microns, and a weaker band at 1.6 microns. The authors fit the first two bands, but not the one at 1.6 microns (~6250 cm-1). Have they considered doing this and comparing the CO2 retrieved between the two spectral regions?*

Yes. The 1.6 micron bands (which are used by TCCON for the whole atmospheric column, > 8 km path) are significantly weaker than the bands near 2.01 and 2.06 microns.  These bands were also fitted, but SNR and scatter was such that, when combined in a SNR-weighted mean with the stronger bands their contribution to the mean is negligible and potentially added additional non-random variability.  We therefore did not include them. We have added the following clarification to the text in section 2.3:

The weaker bands near 6200 cm$^{-1}$ (1.6 $\mu$m) used in total column TCCON analyses were also analysed but are not included because their SNR is much less than that of the stronger 4800-5000 cm$^{-1}$ bands and their contribution to an SNR-weighted mean CO$_2$ retrieval is negligible.

*The authors do not discuss the residuals observed in Figure 4. There are some fits at e.g. 7805 cm-1 in the O2 fit that look like imperfect Voigt fits, but the majority of residuals do not have this appearance. Do the authors have any ideas what causes these large residuals? Do the residuals change as a function of time of day, and could partially result from stray light?*

The residuals are actually already discussed in the extended caption to Table 2, note 7. We have added this text to the main text in section 2.3:

The fibre optic coupling between telescope, source and detector introduces repeatable fringing and interferences in the measured spectra at about 1% of the measured signal intensity.  These structures can be seen in the residual plots of Figure 4 and are quite reproducible over periods of days to weeks. They are larger than the underlying detector noise but much less than the trace gas absorptions, at least for CO$_2$ and O$_2$. Removing or co-fitting an average fibre residual spectrum

during the fit makes only a small (<<1%) difference to the retrieved mole fractions because the fibre residual spectrum is itself derived from the least squares fits to real spectra and is approximately orthogonal to the target gas spectrum.

*The authors use O2 simply as a "system check". Could they use it to filter out spectra that have been influenced significantly by stray light (which then results in the large spikes observed in the O2 time series in Figure 7)?*

As the referee suggests, the CO2 and CH4 data presented have already been filtered for O2>0.225 and time of day between 18:00 and 19:00 which removes almost all the stray-light induced spikes. However this was not noted in the text, and we thank the referee for pointing this out. We have extended the filter to all data spikes where O2>0.225, which reduces the mean CO2 bias factor from 1.025 to 1.024.  We have added to the text to clarify this:

Corresponding spikes are also seen in $CO_2$ and $CH_4$ records and have been filtered to remove data where the retrieved O2 mole fraction is greater than 0.225.

*Do the authors have an explanation for the additional 1.6% O2 bias beyond what is observed by TCCON? If the extra 1.6% discrepancy is due to e.g. unidentified instrumental systematics, could the authors use the O2 to correct the CO2 and CH4 data for the same issues and possibly get better agreement between the open path and point source measurements? The ability to retrieve O2 is one of this instrument's strengths compared to the frequency comb instrument and I think that the authors could capitalize on this strength more than they currently do.*

Although it seems an attractive proposition to scale trace gas amounts to O2 (as is done in TCCON because the atmospheric path is less well known),  there are two problems with this approach. Firstly, scaling the measured mole fractions to O2 adds noise because the scatter  (random noise) in retrieved O2 is similar to that for CO2. Secondly it is not clear that the same quantitative scaling factor would apply to O2 and other trace gases.  Some common sources of error would scale equally for all analysed species, for example pathlength, and pressure and temperature to the extent that they affect calculated air density and hence trace gas mole fractions. These sources of error are discussed and quantified in Table 3 – they are small and do not vary significantly.  While an imperfect instrument function would most likely affect different gases in the same direction, as we see here (all biases are positive), quantitatively the error may be quite different for each species depending for example on linewidths and absorption strengths. In comparing biases to TCCON there are several effects that may cause different biases such as choice of spectral bands,  instrument lineshape, resolution and water vapour interference.
Nevertheless we calculated the CO2 mole fraction for each measurement analogously to TCCON as CO2_column/O2_column*0.2095, which effectively calculates the total air column as O2/0.2095 rather than from pressure and temperature. This predictably changes the scaling factor relative to in situ by 3.6% to -1.1% for CO2 and -0.6% for CH4 (because the O2 mole fraction is 3.6% too high), but also increases the standard deviation in the differences for example for CO2 from 1.1% to 1.7%. An intermediate solution would be to calculate each trace gas mole fraction from trace gas amount, pressure and temperature, but apply a constant overall scaling factor of 1/1.036 to the trace gas

mole fractions based on the 3.6% positive error in O2 retrieval. This partially but not quantitatively offsets the systematically higher concentrations measured with the OP system and appears to improve the agreement and overall scaling factors relative to in situ, but hides the true biases and we prefer to leave the data uncorrected for O2. The following text has been added to section 4.1:

*Open path – in situ bias*

Raw OP measurements are biased high relative to WMO-calibrated in situ measurements at the IUP (western) end of the path, +2.5% for $CO_2$, +3% for CH4 and +3.6% for $O_2$. This assessment relies on the assumption that the atmosphere is well mixed along the open path for windspeeds > 6 m s$^{-1}$ and that there are no actual mole fraction differences under these conditions when the open path and in situ measurements are compared. For comparison, TCCON measurements over much longer atmospheric paths (typically > 10km) have consistent network-wide biases of approximately -3% for $CO_2$, -4.4% for $CH_4$, and +2% for $O_2$. (The TCCON network wide bias for $O_2$ is derived from the comparison of retrieved column $O_2$ amount with atmospheric pressure, and the network wide biases for $X_{CO2}$ (= $CO_2/O_2$*0.2095) and $X_{CH4}$ (= $CH_4/O_2$*0.2095) are -1.0 and -2.4% respectively relative to in situ measurements over the atmospheric column with WMO-scale calibrated analysers (Wunch et al., 2010, updated 2014).) The biases are also similar in magnitude to those seen in mid IR OP and in situ FTIR studies (Smith et al., 2011; Griffith et al., 2012). Thus the observed biases in this study are generally consistent in magnitude with other comparisons of FT spectroscopy with WMO calibrated in situ measurements. As shown in the next paragraph, they are also consistent with an assessment of systematic errors in the retrievals of path-averaged mole fractions from open path infrared spectra.

We have substantially rewritten section 4.1 which discusses errors and biases to address accuracy and precision comments. We have re-quantified the repeatability based on Allan Variance, as suggested by referee 2, re-assessed the biases and compared to those seen in TCCON. The revised text for 4.1 is not reproduced here (> 2 pages, with Figures) but is highlighted in the revised manuscript.

*Do the biases for CO2 and CH4 change if the authors use the same wind cutoff for "well mixed" conditions for both gases? They currently consider CO2 to be well-mixed at wind speeds >6 m/s and CH4 to be well-mixed at wind speeds >2 m/s.*

Yes. We agree in hindsight that there is no justification for having a different windspeed cutoff for CO2 and CH4 and have revised the biases and figures 9-12 accordingly so that both biases are now determined from windspeed > 6 m s$^{-1}$. The tail in the CH4_difference distribution at large differences (due to two events in Aug and Sept) has pushed the CH4 bias up from 0.7% to +3%, and the revised text and discussion in 4.1 and 4.2 reflect these changes.

*I do not agree with the author's assessment that the additional difference between the in-situ and corrected open path CO2 is the result of local emissions. Based on where the City Center is marked in Figure 3, emissions from the city center would affect primarily the open path measurements with S or SE winds as mentioned and would thus bias the open path instrument high relative to the point sensor, not low. Additionally, if there were very local traffic sources from Berliner Strasse, that should*

*be evident as narrow, sharp spikes in the point sensor data (or at the very minimum, the two are likely to agree during most of the night when traffic is presumably minimal and there is no stray light) and that does not seem to be observed in the data*

*in Figure 9. Both time series have similar shapes, but with an offset between the corrected open path data and the in situ data that does seem to drift with time.*

*Do the authors have any idea what is causing the large discrepancy between the open path and in situ CH4 data? Is the power station a natural gas station that would be expected to leak CH4? Are there other local sources of CH4 that are not geographically evenly distributed and might come from the WSW?*

Actual OP – in situ differences are only 2-4 times the measurement precision for both CO2 and CH4, except for two strong positive CH4 events in August and September. It is difficult to make unequivocal conclusions based on such a low "signal to noise". Taking the reassessment of measurement precision and bias into account, as well as the referee's interpretative comments, we have substantially rewritten the interpretation of observed differences in 4.2

**Minor comments:**

*Page 4, lines 13-14: the 3000 spectra that were thrown out due to poor visibility and other weather-related phenomena is approximately 11% of the of the total number of spectra. Why were the other ~30% rejected?*

26000 spectra corresponds to 68% of the available time over the measurement period. Of these 11% were rejected and during the other 21% of the time no spectra were collected due to background spectrum measurement, maintenance or extended bad weather periods. We have revised the text accordingly:

Over the 4 month measurement period more than 26,000 spectra were collected, of which approx. 3000 (11%) were rejected due to poor visibility and low signal or other, normally weather-related effects. In total, taking into account hourly background spectrum measurements, downtime due to maintenance and extended poor weather periods, we collected and analysed usable data for 68% of the total time from 10 July to 4 Nov.

*Page 4, line 25: When was the point sensor last calibrated against the WMO-GAW standards? How often does this instrument need to be calibrated? Could drift or lineshape change in this instrument cause some of the offset between the open-path and point sensor instruments?*
The in situ FTIR analyser calibration strategy is described in detail by Hammer et al (2013) and Vardag et al (2014) – in summary daily target gas measurements and weekly calibration tank measurements to ensure that all measurements are within GAW compatibility goals. We added the sentence:

The calibration frequency (daily target tank, weekly calibration tanks) ensured that all measurements meet GAW compatibility requirements.

*Page 7, line 12: How are the CO2 and CH4 corrected to dry air mole fraction?*

Added:

All raw path averaged mole fractions were converted to dry air using the path-averaged water vapour mole fraction retrieved from the same spectrum:

$$x_{dry} = \frac{x_{wet}}{1 - x_{H2O}}$$

*Page 9, lines 1-2: It is not clear whether the 2.5% and 0.7% biases observed for CO2 and CH4 include the correction or not. Is the total CO2 bias 5%?*

Clarified:

When raw OP measurements are compared to in situ measurements at the IUP (western) end of the path, $CO_2$ and $CH_4$ differences show biases of 2.5% and 0.7% respectively (OP > in situ).

*Figure 2: Please label the launch/receive and retroreflector ends of the path, as well as the location of the point sensor and Berliner Strasse. Please add a North indicator or compass rose. Is there a dominant wind direction that can be added to the plot as well?*
Done, including expanded caption.

*Figure 7: A zoom in on one of the spikes would be nice so the reader can see that they are indeed sharp and centered at ~6 pm local time.*

Done, Insert added to Fig 7, caption amended.

[Figure]

*Figure 11: A zoom in on a time period similar to Figure 9 would be nice.*
Added Figure 11(b).

[Figure]

*Figure 12: y axis in a and x-axis in histogram should be changed to ppb.*
Corrected.

[revised manuscript text omitted]

---

## Author Comment (AC2) · 12 Dec 2017

**AMT-2017-272_AuthorResponse_RC2**

Referee comments are in *italic.*

*Griffith et al present an open-path Fourier transform spectrometer operating in the near infrared with the ability to measure $CO_2$, $CH_4$, $O_2$, $H_2O$ and HDO. Such a system has the potential to be useful for small-scale, continuous greenhouse gas observations. The measurements were performed nearly continuously for 4 months over a 3-km-round-trip path, which is large enough to compare with high-resolution atmospheric models, and reached sensitivities of 1-2 ppm for $CO_2$ and 40 ppb for $CH_4$ with 5-minute measurement times. The authors attempt to quantify the bias and potential systematic errors by comparing to a WMO-calibrated point sensor located at one end of the open path. Overall, the manuscript provides a useful characterization of the system; however, there are some modifications and additional details that I think should be addressed prior to publication.*

We thank both referees for their comments and the consequent improvements to the paper. Responses to general and specific comments are detailed below.

**Major Comments:**
*First, for the precision of the measurements, could the authors please provide an Allan deviation from a well-mixed time period? I find this much easier to understand than just listing the precision in words. Also, why was one-quarter of the peak-to-peak used to estimate the precision instead of taking the standard deviation? Finally, the I found the terminology for variability, precision, uncertainty, and repeatability to be a bit confusing because multiple terms were used interchangeably throughout the manuscript (for example, uncertainty, precision, and repeatability). For clarity, could the authors use a consistent set of terms? I suggest perhaps variability for a time series of data and precision for the final derived result (i.e., remove uncertainty and repeatability).*
The referee's suggestion to use Allan Variance as a measure of repeatability is a good one. The decision to use ¼ of the peak-peak measurements instead of standard deviation was based on (a) the rule of thumb that the peak-peak value of a gaussian distribution for a "reasonable" number of points  is approximately 4-5 times the standard deviation (99.3% of points in a normal distribution lie within5 standard deviations of the mean), and (b) with diurnal variability, there are no periods of time of stable concentrations long enough to adequately calculate standard deviation. However this is an inexact definition and the square root of Allan variance (Allan deviation) is better. We have made and included an extended Allan Variance analysis, and substantially amended the text at  4.1 and Table 2, which has been split into two, with and retrieval sensitivities separate in new Table 3. Note that the repeatability determined from 5 min Allan Variance is significantly lower than the previous estimates in the Discussion paper for $CH_4$ and $O_2$, and unchanged for $CO_2$.

We have reviewed the terms precision, accuracy, uncertainty and variability in the data used in the paper and standardized these to be consistent with terms recommended by the Joint Committee for Guides in Metrology "Evaluation of measurement data — Guide to the expression of uncertainty in measurement (GUM)". According to GUM, "precision" and "accuracy" are non-quantitative terms which should not be assigned numerical values, while terms such as repeatability have quantitative meaning.  In cases where "precision" and "accuracy" have their general, non-quantitative meanings,

we have retained them; where numerical quantities are required we use appropriate terms such as repeatability and bias.

Section 4.1 has been substantially rewritten and is not reproduced here in full (> 2 pages, + tables). It is highlighted in the revised manuscript.

*Could the authors provide some more details about the spectral analysis, specifically: How were the wet measurements converted to dry mole fraction? What polynomial order was used in each spectral region? Was any other filtering or rejection done (e.g., were spectra rejected for anything other than poor signal) or were other corrections applied (e.g., "ghosting" corrections)? Was the ILS measured at multiple times to check for any drift in the ILS?*

For correction to dry air, see response to referee 1. Text added at the top of section 3:
All raw path averaged mole fractions were converted to dry air using the path-averaged water vapour mole fraction retrieved from the same spectrum:

$$x_{dry} = \frac{x_{wet}}{1 - x_{H2O}}$$

The continuum was fitted with a 5-term polynomial, this has been added to the text.

Filtering and rejection was also addressed in the response to referee 1:
26000 spectra corresponds to 68% of the available time over the measurement period. Of these 11% were rejected and during the other 21% of the time no spectra were collected due to background spectrum measurement, maintenance or extended bad weather periods. We have revised the text accordingly:
Over the 4 month measurement period more than 26,000 spectra were collected, of which approx. 3000 (11%) were rejected due to poor visibility and low signal or other, normally weather-related effects. In total, taking into account hourly background spectrum measurements, downtime due to maintenance and extended poor weather periods, we collected and analysed usable data for 68% of the total time from 10 July to 4 Nov.

The ILS was measured on only a few occasions but remained stable.

*The authors note an addition 1.6% $O_2$ bias beyond what is observed in TCCON? Since the authors are using the TCCON spectral database, it does not seem that this added bias would be due to line parameters. Do the authors have a suspected source for this added bias? Is the bias constant in time?*

This comment is covered in detail in the response to referee 1 and the revised section 4.1.

*Have the authors tried fitting $CO_2$ in the 1.6 μm band to compare to the 2 μm region?*
Yes, see response to referee 1:
The 1.6 micron bands (which are used by TCCON for the whole atmospheric column, > 8 km path)

are significantly weaker than the bands near2.01 and 2.06 microns.  These bands were also fitted, but SNR and scatter was such that, when combined in a SNR-weighted mean with the stronger bands their contribution to the mean is negligible and potentially added additional non-random variability. We therefore did not include them. We have added the following clarification to the MS text in section 2.3:

The weaker bands near 6200 $cm^{-1}$ (1.6 μm) used in total column TCCON analyses were also analysed but are not included because their SNR is much less than that of the stronger 4800-5000 $cm^{-1}$ bands and their contribution to an SNR-weighted mean $CO_2$ retrieval is negligible.

*For CH4, do the authors have an explanation for what causes the positive tail in the distribution for >6 m/s wind speeds? This would imply that the open-path measurement reads higher than the point sensor, which seems odd at high wind speeds.*
The positive tail in CH4 record is due mostly to two episodes of high CH4 in the OP data, seen clearly in Figure 10. See also the response to referee 1, who had a similar comment:
Actual OP – in situ differences are only 2-3 times the measurement precision for both CO2 and CH4, except for two strong positive CH4 events in August and September. It is difficult to make unequivocal conclusions based on such a low "signal to noise".
Taking the reassessment of measurement precision and bias into account, as well as the referee's interpretative comments, we have substantially rewritten the interpretation of observed differences in 4.2.

*There is significant structure in the residuals shown in Figure 4 that is not discussed in Section 2.3. Note 7 of Table 2 briefly discusses some fringing in the residuals, which should be covered in Section 2.3. There also seems to be other structure in the residuals such as low-frequency baseline wobble and maybe also structure at the lines. Have the authors looked at the residuals in more detail? How stable are the residuals? What causes the additional structures (they look to be larger than the residuals seen by TCCON)?*
See response to referee 1:
The residuals are actually already discussed in the extended caption to Table 2, note 7. We have added this to the main text in section 2.3:

The fibre optic coupling between telescope, source and detector introduces repeatable fringing and interferences in the measured spectra at about 1% of the measured signal intensity.  These structures can be seen in the residual plots of Figure 4 and are quite reproducible over periods of days to weeks. They are larger than the underlying detector noise but much less than the trace gas absorptions, at least for $CO_2$ and $O_2$. Removing or co-fitting an average fibre residual spectrum during the fit makes only a small (<<1%) difference to the retrieved mole fractions because the fibre residual spectrum is itself derived from the least squares fits to real spectra and is approximately orthogonal to the target gas spectrum.

*Finally, have the authors performed any laboratory measurements, e.g., using a multipass cell, with the system to characterize the bias in that configuration? This would be a useful check to see how the open path compares to a controlled system.*
No, and yes. We did not have a White cell or appropriate optics available for such a test using the same NIR interferometer and transfer optics. In any case, any available cell would be at most of 100m pathlength, which is inappropriate for any quantitative comparison with the 3.2 km open path. Biases due to the optical setup would not be properly assessed by

such a comparison.  The in situ FTIR analyser is such an FTIR-White Cell instrument, but operating in the mid IR with different spectral bands and optics, so again the comparison is not meaningful for the referee's purpose.

**Minor comments:**
*In the introduction, there is no mention of OP-TDLAS or LIDAR systems (e.g., DIAL or IPDA). It would be worthwhile to mention these other technologies and how they compare.*
We have added references to these techniques, however the paper focuses on broad-band-multi species measurements. Following text added to the introduction:
Other recent developments include open path tunable diode laser (TDL) systems (e.g. Dobler et al., 2013; Queisser et al., 2016), and commercially available laser-based open path analysers (e.g. Boreal Laser Inc., Edmonton, Canada). TDL systems are generally applicable only to a single target gas.

*Page 7, line 15: I was under the impression that the WMO scale is not directly SI traceable. Could the authors please clarify where the SI traceability comes from?*
The WMO scales for $CO_2$ and $CH_4$ are traceable to manometric ($CO_2$) or gravimetric ($CH_4$) primary standards generated in the relevant World Calibration Centres (in this case NOAA Boulder labs) and traceable to the standard SI quantities – kilogram, meter etc.  The reference gases used to calibrate the in situ analyser are traceable to these primaries through the ICOS laboratory at MPI-Jena.

*Page 10, line 4: CH4 needs subscript*
Done

*Page 10, line 30: Typo in the units ($cm_{-2}$)*
Done

*Section 4.3: Have you tried over a longer path?*
We tried briefly with measurements to a small retroreflector at approx.. 3 km one-way path. The return beam intensity was too low with this system to achieve useful precision, the weather was not cooperative, and we did not pursue the longer path further.

*Table 1: The values given for the $CO_2$ spectral range do not agree between the wavenumber and micron columns (i.e., 4800 $cm_{-1}$ = 2.08 um).*
Corrected.

*Table 2: in the $dO_2$ column, linestrengths row, what is the "()" for?*
This table has been revised and the () removed

*Fig 7: Perhaps zoom in vertically so that the majority of the time series can be seen.*
Following the suggestion of referee 1 we have added an insert to the figure expanded around one of the spikes. In this insert the reader can also assess the variations in the steady O2 background.  Expanded vertically, the main plot would still appears as "noise" across this wide time range.

[revised manuscript text omitted]

---

## Referee Report (RR1)

The authors have done a good job addressing most of the initial comments from Reviewer 1 and Reviewer 2. However, I have a few comments that still need to be addressed:

Comments:
1. Abstract: in light of the author's determination that they cannot infer any information about local sources and sinks, line 25 in the abstract should be updated.

2. Section 4.1 and Table 2 (Allan Deviation, Precision of measurements): It would be helpful for the authors to present the Allan deviation as a figure rather than a table. If desired, there exists commercial software to calculate these (e.g. Stable32 and presumably others). Additionally, it would be helpful for the authors to restrict their Allan analysis to time periods when the $CO_2$ and $CH_4$ is well-mixed. The purpose of the Allan deviation is to provide information about the instrument (i.e. is it dominated by white nose or colored noise? Where does it flatten out and start increasing? Based on the time scale, can the factor that sets that turnaround point be determined?) rather than the long-term atmospheric variability.

3. Section 4.1, Open path – in situ bias: The authors quote two offsets for TCCON $CO_2$ and $CH_4$. One is listed as the "network-wide bias" and the other is "network-wide bias of $X_{CO2}$ and $X_{CH4}$". Please clarify the difference between these two biases.

4. Section 4.2 $CH_4$ and Figure 11, especially the insert: The authors provide no explanation for the discrepancy between the OP and in situ instrument. The in situ instrument seems to show a diurnal cycle of about 50 ppb for $CH_4$, but the OP instrument seems to wander all over the place. Sometimes it is higher than the OP instrument (e.g. time periods on 21 Aug. and 22 Aug.) and sometimes it is significantly lower (e.g. 14-17 Aug.), but it does not seem to show any sort of trend or correlation with the OP instrument. This is in significant contrast to $CO_2$ and Figure 9 (especially inset) where it is clear that there is a tight correlation between the two instruments but an offset between them. The authors need to provide some discussion of this. Is the $CH_4$ spectral region affected more strongly by stray light than the $CO_2$ spectral region and variations in stray light could be causing this? (Table 3 seems to suggest that there is indeed an enhanced stray light effect.) Does the OP instrument light path cross anything that might be a $CH_4$ source (or sink) that would disperse by the time it reaches the in-situ instrument? The largest differences seem to occur when the wind is out of the SE (according to Figure 12b) but there is also a tight correlation at 330 degrees on ~1 Aug. time period (again according to Figure 12b). Based on Figure 4, the $H_2O$ interference in the $CH_4$ retrieval window seems to be quite strong. Does the discrepancy between the OP/in-situ instrument correlate with water concentration or relative humidity?

5. Section 4.2 regarding the diurnal offsets: In addition to the temperature possibly causing diurnal offsets, it seems that stray light should also have a diurnal cycle. Have the authors tried correlating the (OP – in situ) quantity that varies diurnally with e.g. $O_2$ enhancement, or some other measure of stray light? (This would of course not explain the wind dependence observed for $CO_2$ though.)

Editorial comments:

1. Figure 7: the inset only has one tick mark labeled in the updated figure so it is not possible to tell the time scale of the spike width. Please update to include at least one other tick label.

2. Section 4.2: change to "For corrected $CO_2$…" and "For corrected $CH_4$…" to clarify that these are the resulting offsets after the high windspeed correction.

3. The authors should double check their manuscript for subscript errors.

4. The authors should check their references for superscript and subscript errors.

5. Table 4: The authors should clarify that this refers to their *uncorrected* offsets.

6. Figure 4: It would be helpful for the authors to also plot the other species that are retrieved in each window.

---

## Author Response (AR2)

**Revised manuscript Dec 2017**

**Author response to 2 reviewers, 19 Jan 2018**

We thank both reviewers for their constructive comments and submit the following responses. Reviewer comments are in *italic*, extracted changes to the manuscript are in red.

**Reviewer # 1**

*Comments:*

*1.Abstract: in light of the author's determination that they cannot infer any information about local sources and sinks, line 25 in the abstract should be updated.*

Done. New text at line 25:
We observe significant differences of the order of a few ppm for $CO_2$ and a few tens of ppb for $CH_4$ between the open path and point measurements 2-4 times the measurement repeatability, but we cannot assign the differences to specific local sources or sinks.

*2. Section 4.1 and Table 2 (Allan Deviation, Precision of measurements): It would be helpful for the authors to present the Allan deviation as a figure rather than a table. If desired, there exists commercial software to calculate these (e.g. Stable32 and presumably others). Additionally, it would be helpful for the authors to restrict their Allan analysis to time periods when the $CO_2$ and $CH_4$ is well-mixed. The purpose of the Allan deviation is to provide information about the instrument (i.e. is it dominated by white nose or colored noise? Where does it flatten out and start increasing? Based on the time scale, can the factor that sets that turnaround point be determined?) rather than the long-term atmospheric variability.*

We have added Figure 13 to show Allan Deviations graphically but also kept and extended the tabulated values in Table 2. It is not practicable to restrict the analysis to only "well mixed" time periods because (1) they are short and discontinuous across 4 months, usually only a few hours on windy days, and (2) the selection of what is well mixed is subjective. One option would be to calculate Allan deviations from only the data with windspeed > 6 m/s, but there are insufficient data for this to be useful. We do accept the validity of this comment and we had already considered the separation of instrument and atmospheric variability; in the existing text we presented the ADs after subtracting a smoothed curve drawn through the data to remove the gross atmospheric variability. We have added the explicit smoothed-subtracted results to table 2 and figure 13. The revised first paragraph of 4.1 is now:

Table 2 and Figure 13 show Allan deviations (AD, the square root of Allan Variance (Werle et al., 1993)) for open path and in situ $CO_2$, $CH_4$ and $O_2$ measurements and the open path – in situ differences. The ADs in Table 2 were calculated from the period 11 Aug 06:00 - 27 Aug 18:00 when diurnal variation was minimal and short term repeatability can be best estimated; they are presented for 5 min (single measurements), 1 hour and 6 hour averaging times. The 5 minute ADs for the raw data provide upper limits for the instrument or measurement noise, since the variability is dominated by instrument noise but there is also the possibility of a small contribution from atmospheric variability over 5 min time scales. For comparison, a smoothed curve through the raw

data was subtracted from the raw data to remove the gross atmospheric variation (2$^{nd}$ order Savitzky-Golay smoothing, 15 points, approx 1-hour smoothing) and ADs recalculated ("smoothed-subtracted" data). Five-minute ADs and the standard deviations of the smoothed-subtracted data are similar to those of the raw data at 5 min and are also shown in Table 2 and Figure 13; the smoothed-subtracted ADs decrease with averaging time out to 6 hours approximately as expected for random noise. The 5 min Allan deviation values are ~1.7 ppm (0.4%) for $CO_2$, 23 ppb (1.2%) for $CH_4$ and 0.0016 (0.7%) for $O_2$. For in situ measurements they are lower, reflecting the better repeatability of the in situ analyser: 0.63 ppm (0.15%) for $CO_2$ and 2.1 ppb (0.1%)  for $CH_4$.  We take these values as our best estimates of the 1-$\sigma$ repeatability of the measurements due to the instrument noise with minimum influence from atmospheric variability.

*3. Section 4.1, Open path – in situ bias: The authors quote two offsets for TCCON $CO_2$ and $CH_4$. One is listed as the "network-wide bias" and the other is "network-wide bias of $X_{CO2}$ and $X_{CH4}$". Please clarify the difference between these two biases.*

Clarified:
For comparison, TCCON measurements of total columns over much longer atmospheric paths (typically > 10km) have consistent network-wide biases of approximately -3% for $CO_2$, -4.4% for $CH_4$, and +2% for $O_2$. (The TCCON network wide bias for $O_2$ is derived from the comparison of retrieved column $O_2$ amount with atmospheric pressure, and the network wide biases for $X_{CO2}$ (= $CO_2/O_2*0.2095$) and $X_{CH4}$ (= $CH_4/O_2*0.2095$), which include and partially cancel the biases in both target species and $O_2$, are -1.0 and -2.4% respectively relative to in situ measurements over the atmospheric column with WMO-scale calibrated analysers (Wunch et al., 2010, updated 2014).)

*4. Section 4.2 $CH_4$ and Figure 11, especially the insert: The authors provide no explanation for the discrepancy between the OP and in situ instrument. The in situ instrument seems to show a diurnal cycle of about 50 ppb for $CH_4$, but the OP instrument seems to wander all over the place. Sometimes it is higher than the OP instrument (e.g. time periods on 21 Aug. and 22 Aug.) and sometimes it is significantly lower (e.g. 14-17 Aug.), but it does not seem to show any sort of trend or correlation with the OP instrument. This is in significant contrast to $CO_2$ and Figure 9 (especially inset) where it is clear that there is a tight correlation between the two instruments but an offset between them. The authors need to provide some discussion of this. Is the $CH_4$ spectral region affected more strongly by stray light than the $CO_2$ spectral region and variations in stray light could be causing this? (Table 3 seems to suggest that there is indeed an enhanced stray light effect.) Does the OP instrument light path cross anything that might be a $CH_4$ source (or sink) that would disperse by the time it reaches the in-situ instrument? The largest differences seem to occur when the wind is out of the SE (according to Figure 12b) but there is also a tight correlation at 330 degrees on ~1 Aug. time period (again according to Figure 12b). Based on Figure 4, the $H_2O$ interference in the $CH_4$ retrieval window seems to be quite strong. Does the discrepancy between the OP/in-situ instrument correlate with water concentration or relative humidity?*

Unfortunately the OP $CH_4$ measurements lack the precision and stability to clearly show the observed differences with in situ measurements. We find that discussion of the differences can only be speculative, and did not therefore discuss in detail. There is no correlation between the differences and water vapour concentrations, excluding a spectroscopic cross-talk explanation. Below is a quick plot of CH4 differences (in ppb) vs H2O amounts (in %) – although the higher CH4 differences occur at higher water vapour concentrations, there is no correlation per se, the high water levels may simply be temperature or seasonally related.

[Figure]

To address the reviewer's concerns, we have added the following to the CH4 section in 4.2:

The observed differences in OP $CH_4$ relative to in situ measurements are only marginally greater than the OP measurement stability and repeatability and are difficult to quantify or assign with any certainty to specific atmospheric conditions or local sources or sinks. $CH_4$ relative precision is lower than for $CO_2$ because of both the absolute strength of the $CH_4$ absorption features and their strength relative to overlapping water vapour absorption (Figure 4). There is no correlation between the OP-in situ differences and coincident water vapour amounts derived from the same spectra, suggesting that the $CH_4$ differences are not an artefact due to spectra overlap. There are numerous possible small, local point sources, such as natural gas or wastewater piping leaks, that may affect the observed differences, but with this level of precision, detailed interpretation can only be speculative.

*5. Section 4.2 regarding the diurnal offsets: In addition to the temperature possibly causing diurnal offsets, it seems that stray light should also have a diurnal cycle. Have the authors tried correlating the (OP – in situ) quantity that varies diurnally with e.g. O2 enhancement, or some other measure of stray light? (This would of course not explain the wind dependence observed for CO2 though.)*

The effect of stray light is discussed in 2.3 – after removing the direct evening peaks, residual stray light effects are small, < 1-2 ppm for CO2 compared to an instrument measurement precision of 1.7 ppm. Any correlation would be difficult to identify beneath the instrument noise level, especially given the large natural diurnal variation in CO2. Indeed there is no significant correlation observable between for example CO2 differences and O2.

*Editorial comments:*
1.  *Figure 7: the inset only has one tick mark labeled in the updated figure so it is not possible to tell the time scale of the spike width. Please update to include at least one other tick label.*

Done

2.  *Section 4.2: change to "For corrected CO2..." and "For corrected CH4..." to clarify that these are the resulting offsets after the high windspeed correction.*

Done

3. *The authors should double check their manuscript for subscript errors.*

Done - will double check on the final clean copy after acceptance.

4. *The authors should check their references for superscript and subscript errors.*

Done.

5. *Table 4: The authors should clarify that this refers to their \*uncorrected\* offsets.*

Done

6. *Figure 4: It would be helpful for the authors to also plot the other species that are retrieved in each window.*

In all windows the remaining features are dominated by water vapour – we think adding water vapour to the plot would make the message less clear than currently. We have amended the caption to make this point. The other interfering species are listed in Table 1.

***Reviewer #2***

*First, is there a reason that the Allan deviation is provided in a table instead of in a figure? It would be nice to see the variation with averaging time for a variety of times.*

See response to reviewer 1, who made the same request. Allan plots have been added and the discussion enhanced.

*Second, would it be possible to include the wind speed in the spectrum time series?*

The differences are plotted against  windspeed and wind direction coloured by time  in Figures 10 and 12.  As can be seen there, there is very little correlation, and it is also difficult to visualise any correlation when the data are co-plottedas time series such as in figures 9 and 11. Adding windspeed adds complexity without bringing any new message to the plot, and we prefer not to include it.

*Finally, could the authors provide the instrument noise level. This could be, e.g., the RMS of residuals (or in a spectral region with no absorption) after removal of the fiber residual spectrum.*

Good idea. Added text to section :

[revised manuscript text omitted]